# 3D reconstruction of neuronal allometry and neuromuscular projections in asexual planarians using expansion tiling light sheet microscopy

Jing Lu[1,2†], Hao Xu[1,3,4†], Dongyue Wang[2,5], Yanlu Chen[2,5], Takeshi Inoue[6], Liang Gao[2,3,5]*, Kai Lei[3,4,5]*

[1]College of Life Sciences, Zhejiang University, Hangzhou, China; [2]Key Laboratory of Structural Biology of Zhejiang Province, School of Life Sciences, Westlake University, Hangzhou, China; [3]Westlake Laboratory of Life Sciences and Biomedicine, Hangzhou, China; [4]Key Laboratory of Growth Regulation and Translational Research of Zhejiang Province, School of Life Sciences, Westlake University, Hangzhou, China; [5]Institute of Biology, Westlake Institute for Advanced Study, Hangzhou, China; [6]Division of Adaptation Physiology, Faculty of Medicine, Tottori University, Yonago, Japan

*For correspondence:
gaoliang@westlake.edu.cn (LG);
leikai@westlake.edu.cn (KL)

†These authors contributed equally to this work

Competing interest: The authors declare that no competing interests exist.

## eLife Assessment

Lu and colleagues developed an **important** imaging protocol that combines expansion microscopy, light-sheet microscopy, and image segmentation for use with the planarian *Schmidtea mediterranea*, a powerful model system for regeneration. This represents a substantial improvement on current standards and enables more rapid data acquisition. The utility of this **solid** protocol is demonstrated by quantifying several aspects of this flatworm's neural anatomy and musculature during homeostasis and regeneration. This work will be of interest to researchers looking to implement more systematic approaches towards imaging and quantifying intact specimens.

**Abstract** The intricate coordination of the neural network in planarian growth and regeneration has remained largely unrevealed, partly due to the challenges of imaging the CNS in three dimensions (3D) with high resolution and within a reasonable timeframe. To address this gap in systematic imaging of the CNS in planarians, we adopted high-resolution, nanoscale imaging by combining tissue expansion and tiling light-sheet microscopy, achieving up to fourfold linear expansion. Using an automatic 3D cell segmentation pipeline, we quantitatively profiled neurons and muscle fibers at the single-cell level in over 400 wild-type planarians during homeostasis and regeneration. We validated previous observations of neuronal cell number changes and muscle fiber distribution. We found that the increase in neuron cell number tends to lag behind the rapid expansion of somatic cells during the later phase of homeostasis. By imaging the planarian with up to 120 nm resolution, we also observed distinct muscle distribution patterns at the anterior and posterior poles. Furthermore, we investigated the effects of *β-catenin-1* RNAi on muscle fiber distribution at the posterior pole, consistent with changes in anterior-posterior polarity. The glial cells were observed to be close in contact with dorsal-ventral muscle fibers. Finally, we observed disruptions in neural-muscular networks in *inr-1* RNAi planarians. These findings provide insights into the detailed structure and potential functions of the neural-muscular system in planarians and highlight the accessibility of our imaging tool in unraveling the biological functions underlying their diverse phenotypes and behaviors.

## Introduction

The CNS stands as a marvel of intricate organization, enabling the execution of complex functions crucial for an organism's survival (*Cajal, 1995*). It is the hub for processing and coordinating information throughout the body, employing specialized regions with distinct structures and functions (*Bullock and Horridge, 1965*). However, the regenerative capacity of the CNS poses a formidable challenge, as it exhibits limited ability for de novo regeneration (*Obernier et al., 2014*).

The planarian CNS is a fascinating model for studying neural regeneration (*Agata et al., 1998*). Planarians are flatworms that possess a relatively simple CNS, yet they have an impressive ability to regenerate their neural tissue. The planarian CNS is organized into different molecular and functional domains defined by the expression of specific neural genes (*Cebrià et al., 2002b*). Planarians can regenerate functional brains from even tiny body fragments, highlighting their remarkable regenerative capabilities (*Umesono and Agata, 2009*). This unique regenerative potential is attributed to the presence of pluripotent stem cells called neoblasts, which can differentiate into various cell types, including neurons (*Cebrià, 2007*). The availability of hundreds of genes expressed in planarian neurons, coupled with the ability to silence them through RNA interference, has facilitated the unraveling of the molecular mechanisms underlying CNS regeneration in these organisms (*Cebrià, 2007*). The study of planarian CNS regeneration provides valuable insights into the fundamental processes of neural regeneration, which may have implications for regenerative medicine and understanding human nervous system repair.

Understanding the mechanisms underlying CNS regeneration requires applying powerful tools to study its structure and organization at the cellular to sub-cellular levels. Advanced imaging techniques, including high-resolution microscopy, offer exceptional opportunities to gain invaluable insights into the intricate architecture of the CNS. Gained from advanced imaging techniques, researchers can harness knowledge from the regenerative wonders observed in nature that hold promise for promoting CNS regeneration (*Dodt et al., 2007*; *Tomer et al., 2011*). However, the intricate network and the dynamics of planarian CNS have remained largely unrevealed due to the challenges of imaging the CNS in 3D with high resolution within a reasonable timeframe.

Tiling light sheet microscopy (TLSM) is a flexible imaging technique that has been adapted for use in live organisms and cleared tissues (*Gao, 2015*; *Fu et al., 2016*). Its flexible multicolor 3D imaging ability has been shown across a variety of samples, from structures as complex as the mouse spinal cord to the intricate tissues of planarians (*Chen et al., 2020*; *Xie et al., 2023*). In TLSM, a thin and focused light sheet is used to illuminate the sample from the side, exciting fluorophores close to the focal plane. By tiling the light sheet within the imaging field of view at multiple positions and using the images generated by the thinnest section of the light sheet, researchers can create a comprehensive and high-resolution image of the entire sample. This method combines the benefits of light sheet microscopy, which offers high spatial resolution and imaging speed, with tiling capabilities to capture larger samples (*Chen et al., 2020*). This technique is particularly useful for imaging cleared tissues, enabling rapid multicolor 3D imaging with micron-scale to submicron-scale spatial resolution (*Chen et al., 2020*). Expansion microscopy has been employed in planarian studies for the detailed visualization of neuronal structures (*Wang et al., 2016*; *Khariton et al., 2020*). It remained a challenge to image the entire CNS in 3D at high resolution within a reasonable time frame. While tissue clearing is a common practice in microscopy, we found it particularly useful as a pre-expansion treatment for lipid-rich samples such as planarians. This process allows homogenization without the need for heating or proteinase treatment. Clearing and Magnification Analysis of Proteome (C-MAP) was able to preserve the natural proteins during expansion, which allows the use of conventional FISH and antibody staining (*Chen et al., 2020*). The combination of C-MAP and tiling light sheet microscopy has achieved improved 3D resolution, signal-to-noise ratio, and sample compatibility (*Chen et al., 2015*; *Ku et al., 2016*; *Tillberg et al., 2016*; *Chang et al., 2017*; *Gao et al., 2019*; *Wassie et al., 2019*). TLSM has greatly advanced our understanding of complex biological systems and has opened new possibilities for studying cellular dynamics and interactions within multicellular organisms (*Gao, 2015*; *Fu et al., 2016*). The combination of TLSM and C-MAP suggests a potential method to study the regenerative CNS in planarian and other non-traditional model organisms.

In this study, we applied TLSM and C-MAP to record the planarian spatial information at single cellular or higher resolution levels. We present a 3D tissue reconstruction method to investigate neuron type diversity and development at the single-cell level by labeling various neuron types,

including cholinergic, GABAergic, octopaminergic, dopaminergic, and serotonergic neurons. We successfully quantitatively profiled neurons at the single-cell level in over 400 wild-type planarians during homeostasis and regeneration. In addition to obtaining higher resolution images of known structures within planarians, such as muscles, we also discovered previously unreported muscle-muscle and neuron-muscle connections. We further provided evidence that suggests muscle fibers as a scaffold for targeted neuron projection. These results are of significant interest as they contribute to our understanding of how the primitive CNS coordinates the behavior and the underlying mechanism involved in the precise regeneration of neurons and their networks.

## Results

### Establishment of 3D tissue reconstruction using expansion tiling light sheet microscopy for planarian *Schmidtea mediterranea*

We first set up the experiment pipeline utilizing Clearing and Magnification Analysis of Proteome (C-MAP) for planarian expansion and tiling light-sheet microscope (TLSM) for imaging (*Figure 1A*). The expansion procedure was performed after the conventional staining in planarians (*Chen et al., 2020*). To improve the efficiency of sample processing, we have made several modifications to the original protocol (*Figure 1A*). First, we incorporated tissue clearing to ensure uniform homogenization of the entire planarian. Second, instead of relying on the conventional gelation incubation at 37 °C, we expedited gelation by exposing the samples to violet light for 30 s. Third, we conducted the procedure on ice to minimize the impact of high-temperature gelation. Last, we reduced or eliminated the time required for tissue clearing for smaller samples. It is important to point out that the strength of our C-MAP protocol lies in its fluorescence-protective nature and user convenience. Notably, the sample can be expanded up to 4.5-fold linearly without the need for heating or proteinase digestion, which helps preserve fluorescence signals. In addition, the entire expansion process can be completed within 48 hr. Based on our research requirement, two spatial resolutions were adopted to image expanded planarians, $2 \times 2 \times 5$ µm$^3$ and $0.5 \times 0.5 \times 1.6$ µm$^3$. The resolution can be further improved to 500 nm and 120 nm, respectively. The total hours required for expansion and imaging were summarized (*Figure 1—figure supplement 1A*). In the case of a 2 mm planarian, imaging at $2 \times 2 \times 5$ µm$^3$ spatial resolution requires approximately 1 hr with dual channel imaging. Imaging at $0.5 \times 0.5 \times 1.6$ µm$^3$ resolution requires about 12 hr. While our current analysis focused on cellular-level structures, our method can achieve a resolution of $0.5 \times 0.5 \times 1.6$ µm$^3$ and a spatial resolution of $0.12 \times 0.12 \times 0.4$ µm$^3$ with a 4.5×isotropic expansion, which is comparable to previously reported methods (*Fan et al., 2021*; *Wang et al., 2016*). The individual images were able to be conveniently integrated into a 3D tiff. file (*Figure 1—figure supplement 1B*). After all, we believe it is a practical pipeline to image planarians in 3D with high resolution within an acceptable time frame.

We next set up the pipeline of 3D tissue reconstruction and cell segmentation for planarian CNS. To accurately count individual cells, we developed an automatic cell-counting pipeline to segment various planarian tissues and individual cells, including a plane section of the head, a layer of epidermis, and the whole organ of the pharynx (*Figure 1—figure supplement 1C–P*). This pipeline detects nucleus boundaries and assigns labels to each nucleus, thus facilitating accurate cell counting (*Figure 1—figure supplement 1C–G, Q-T*, *Figure 1—video 1*). Additionally, the neuron system of *S. mediterranea* is complex and characterized by considerable diversity among glutamatergic, glycinergic, and peptidergic neurons in planarians and many neurons in *S. mediterranea* express more than one neurotransmitter or neuropeptide, which adds further complexity to the system (*Cebrià et al., 2002a*; *Collins et al., 2010*; *Fraguas et al., 2012*; *Ong et al., 2016*; *Rawls et al., 2009*; *Shimoyama et al., 2016*; *Vaaga et al., 2014*; *Wyss et al., 2022*). We used five markers for a proof of concept illustration. By employing Fluorescence in Situ Hybridization (FISH), we successfully visualized a variety of planarian neurons, including cholinergic (*chat$^+$*), serotonergic (*tph$^+$*), octopaminergic (*tbh$^+$*), GABAergic (*gad$^+$*), and dopaminergic (*th$^+$*) neurons based on their well-characterized roles in planarian neurobiology and the availability of reliable markers. (*Figure 1—figure supplement 2A*, *Figure 1—video 2*; *Nishimura et al., 2007*; *Currie et al., 2016*). The combination of these five types of neurons constitutes a neuron pool that enables the labeling of most of the neurons throughout the entire body, including the eyes, brain, and pharynx (*Figure 1B*). Segmentation of each neuron type showed their spatial atlas in the head (*Figure 1—figure supplement 2B*). Similarly, the *estrella$^+$* glial cells can be

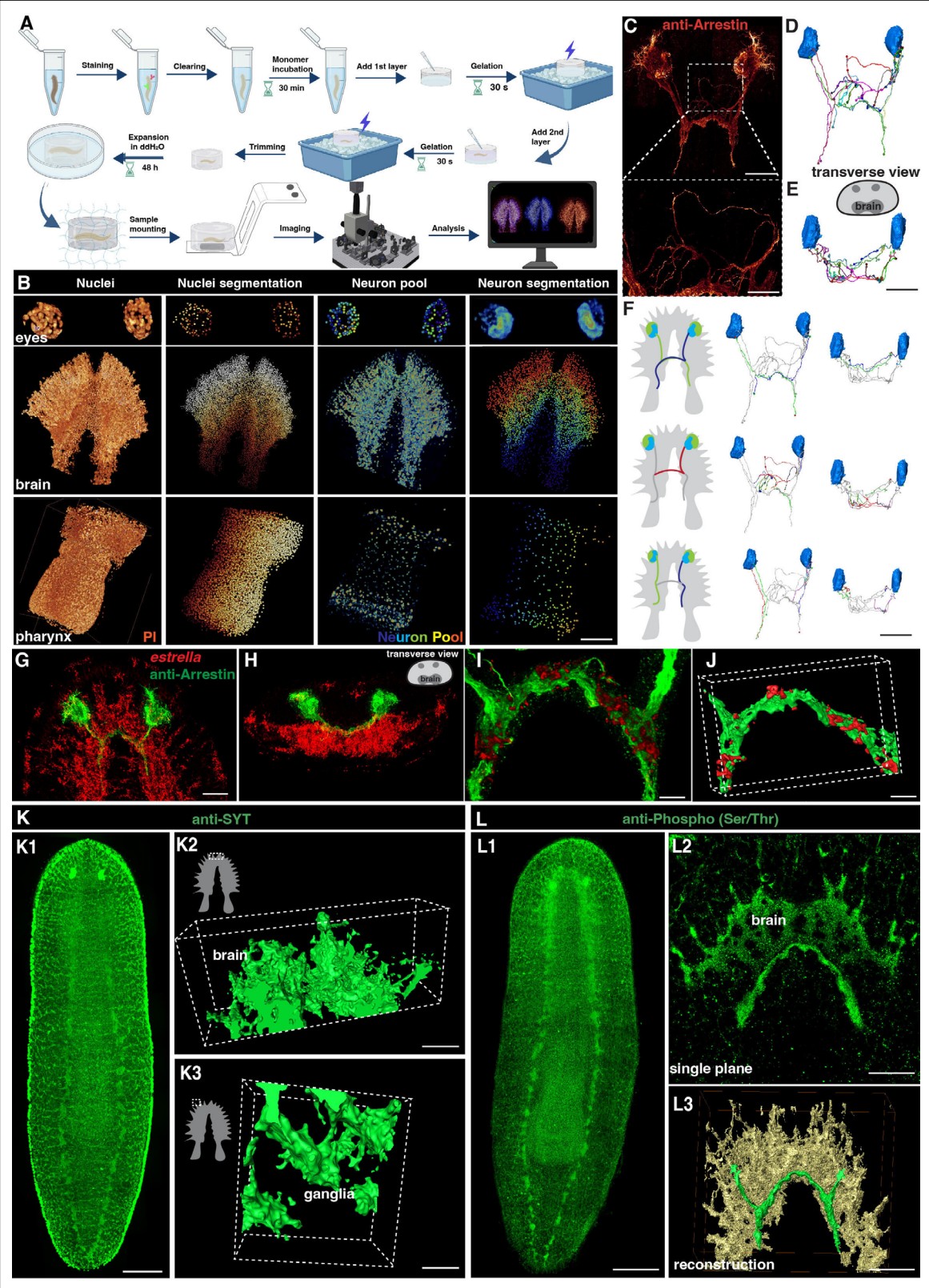

**Figure 1.** Tiling light sheet microscopy for imaging planarian central nervous system (CNS). (**A**) Planarian expansion workflow. Planarians were fixed and stained with FISH or immunostaining, followed by tissue expansion and tiling light sheet microscopy imaging. Created with Biorender. (**B**) Segmentation of PI and neuron pool riboprobes labeled cells in eyes, brain, and pharynx. Scale bar, 600 μm. (**C**) Staining of anti-Arrestin antibody for the planarian visual system. Scale bar, 600 μm. The lower image shows a magnification of the selected area in the upper image. Scale bar, 200 μm. (**D**) Neuron tracing

*Figure 1 continued on next page*

*Figure 1 continued*

of the upper image in panel C. (**E**) Tracing of single neurons in transverse view. Scale bar, 600 µm. (**F**) Traced axon projection trajectories from each eye. Scale bar, 600 µm. (**G**) Dual staining of glial cells (*estrella*⁺) and visual system (anti-Arrestin⁺) in the head region of a wild-type planarian. Scale bar, 350 µm. (**H**) Xz view of the image in panel G. (**I**) Zoom in of panel G shows the glial cells close to the visual axons. Scale bar, 100 µm. (**J**) Reconstruction of the glial and visual system. Scale bar, 100 µm. (**K1**) Anti-SYT staining in a wild-type planarian. Scale bar, 600 µm. (**K2**) The 3D reconstructed image of ganglia in the anterior tip of the brain. Scale bar, 120 µm. (**K3**) The 3D reconstructed image of ganglia in the branch region of the brain. Scale bar, 30 µm. (**L1**) Anti-Phospho (Ser/Thr) staining in a wild-type planarian. Scale bar, 600 µm. (**L2**) Single plane image of the brain. Scale bar, 150 µm. (**L3**) 3D reconstruction of brain region. Green represents the visual neurons, and yellow represents the brain. Scale bar, 180 µm.

The online version of this article includes the following video, source data, and figure supplement(s) for figure 1:

**Figure supplement 1.** TLSM imaging and cell segmentation for planarian pharynx, brain, and epidermis.

**Figure supplement 1—source data 1.** Source data for *Figure 1—figure supplement 1*.

**Figure supplement 2.** Five major neuron types in planarians.

**Figure 1—video 1.** Examples of cell segmentation in planarians after ETLSM.

https://elifesciences.org/articles/101103/figures#fig1video1

**Figure 1—video 2.** 3D images of the five major neuron types in planarians.

https://elifesciences.org/articles/101103/figures#fig1video2

**Figure 1—video 3.** 3D image of the visual projections in a wild-type planarian.

https://elifesciences.org/articles/101103/figures#fig1video3

visualized and segmented (*Figure 1—figure supplement 2C*). The segmentation pipeline applied at 160 nm resolution at the single cell level was achieved for the nucleus and the cell body of neurons.

To visualize the neural network, we further stained the anti-Arrestin to image the visual projections (*Figure 1C–E*, *Figure 1—video 3*). We traced the trajectories of the photoreceptor axons, corroborating the existence of both ipsilateral and contralateral projections (*Figure 1F*; *Okamoto et al., 2005*). Photoreceptor axons displayed the trajectories to the contralateral or the ipsilateral side of the brain. Choice points were observed at the optic chiasm, consistent with the previous description (*Agata et al., 1998*; *Okamoto et al., 2005*; *Scimone et al., 2020*). Glial cells have been observed to be closely associated with neurons in the brain region (*Wang et al., 2016*; *Roberts-Galbraith et al., 2016*). Additionally, it has been reported that glial cells might assist in the projection of photoreceptors (*Chandra et al., 2023*). To validate these observations, we performed co-staining of anti-Arrestin and *estrella* (*Figure 1G and H*). Our results consistently showed a strong association between glial cells and the projections of photoreceptors in the brain region (*Figure 1I and J*). To visualize the neuronal network of the planarian, we used antibody staining with anti-SYT (*Figure 1K*; *Tazaki et al., 1999*) and anti-Phospho (Ser/Thr; *Figure 1L*), respectively. Both anti-SYT and anti-Phospho (Ser/Thr) staining effectively stained the planarian brain and ventral nerve cord (VNC), therefore facilitating the observation of the planarian neuron network. Above all, we developed a platform for digital documentation and exploration of planarian CNS structures.

## Cell counting reveals a potential threshold in the increase of neuron numbers during planarian growth

Our method allows for a comprehensive quantitative analysis of the cell number change. Planarians ranging in length from 1 mm to 10 mm were carefully selected during the homeostatic phase to model planarian growth (*Figure 2A*, *Figure 2—figure supplement 1A*). In total, 99 samples were analyzed for 3D tissue reconstruction and cell segmentation. Images of alive planarians were captured, and accurate length was measured. By dual staining of the neuron pool and propidium iodide (PI), 3D images of the planarians were analyzed to measure the volume, length, width, and depth of all planarians, and numbers of whole-body cells and neurons (*Figure 2B and C*). The volume and surface areas were quantified, revealing a consistent ratio of the square root of surface area to the cube root of volume during homeostasis (*Figure 2—figure supplement 1B*). The cell number-to-volume ratio remained stable in planarians during homeostasis (*Figure 2—figure supplement 1C*). Furthermore, brain volumes were measured, and brain volume increases proportionally with the growth of body length and volume (*Figure 2—figure supplement 1D, E*). Our results indicate the ability of planarians to flexibly regulate their cell number and scale of surface area relative to volume to adapt to the developmental changes during homeostasis.

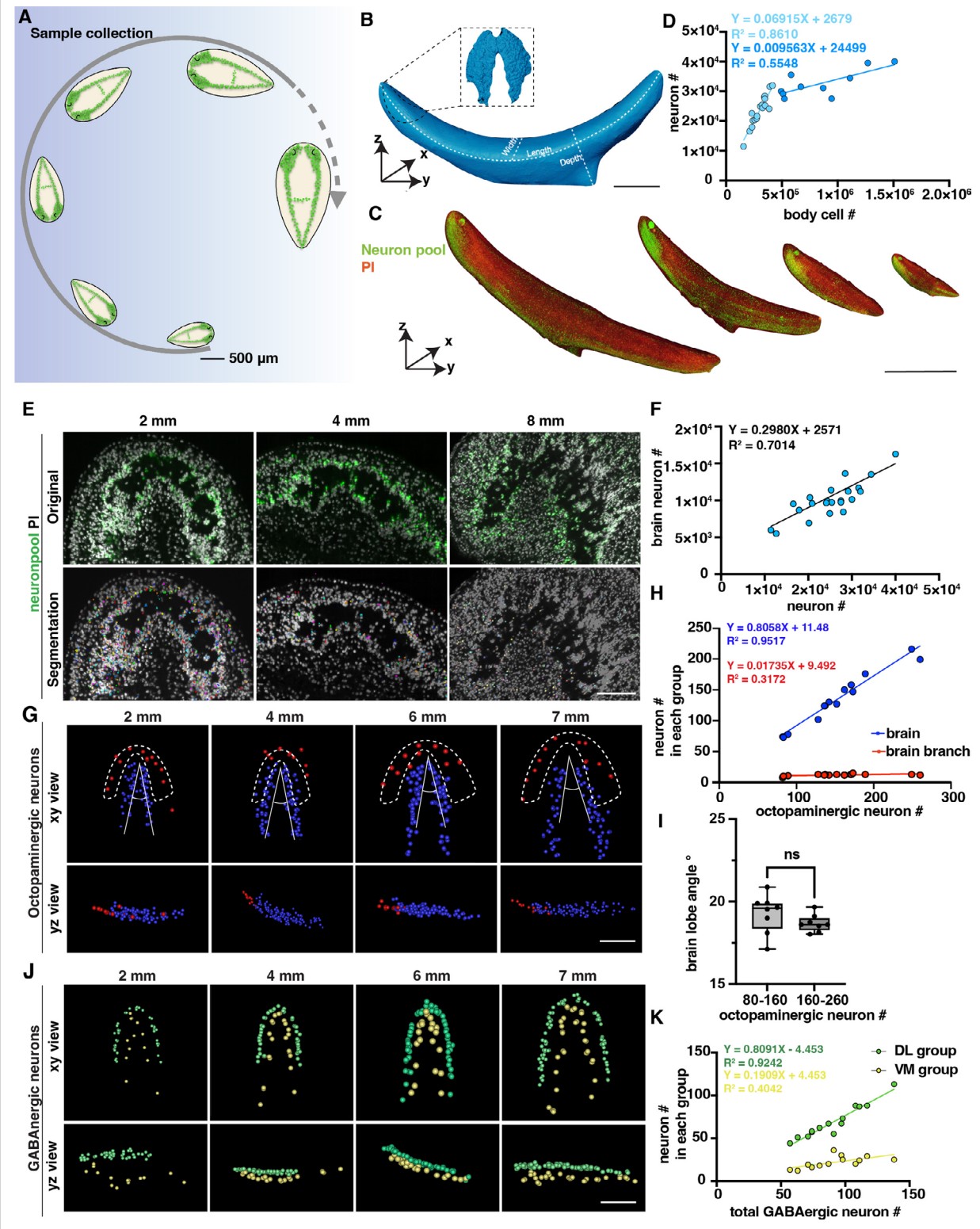

**Figure 2.** Changes in the number of planarian neurons during homeostasis. (**A**) Workflow of sample collection of homeostatic planarians. Scale bar, 500 µm. (**B**) 3D reconstruction of a planarian with PI staining. Length, volume, and surface area were measured using reconstructed images. The planarian brain is segmented and shown in a black dotted box. Scale bar, 2700 µm. (**C**) Representative fluorescent images of planarians stained with neuron pool riboprobes and PI at sizes of 2 mm, 4 mm, 6 mm, and 8 mm. Scale bar, 6000 µm. (**D**) Dot plot shows the correlation of whole-body cell number with neuron number in different sizes of intact planarians during homeostasis. Two trendlines are shown to represent planarians with differing

*Figure 2 continued*

cell counts. (**E**) Zoomed grayscale (PI) and segmented neurons (green) of selected brain regions of planarians at sizes of 2 mm, 4 mm, and 8 mm. Scale bar, 450 μm. (**F**) Correlation of total neuron number with neuron number in brains in different sizes of intact planarians. (**G**) Segmented images of octopaminergic neurons in the main brain region (blue) and brain branch region (red) of planarians with the indicated body length are shown with xy and yz views. Scale bar, 900 μm. (**H**) Dot plot shows the correlation of the total number of octopaminergic neurons with the octopaminergic neuron number in the brains (blue dots) and branches (red dots). (**I**) The plot illustrates the correlation between the angle of the brain lobe and the number of octopaminergic neurons in two groups of intact planarians: one with 80–160 octopaminergic neurons and the other with 160–220 octopaminergic neurons. n=8 in each condition. The data is shown as the mean ± SEM. Statistical significance was evaluated using the two-tailed unpaired Student's t-test, with **$p<0.01$, ***$p<0.001$ indicating significance, while ns indicates lack of significance. (**J**) Segmented images of GABAergic neurons in the brains of planarians with the indicated body length are shown with xy and yz views. Scale bar, 900 μm. (**K**) Dot plot shows the correlation of the total number of GABAergic neurons with the GABAergic neuron number in the VM region (yellow dots) and DL region (green dots).

The online version of this article includes the following source data and figure supplement(s) for figure 2:

**Source data 1.** Source data for *Figure 2*.

**Figure supplement 1.** Neuron changes during homeostasis.

**Figure supplement 1—source data 1.** Source data for *Figure 2—figure supplement 1*.

Previous studies reported that body cells increase in number in correlation with planarian size growth through quantitative western blotting of worm lysates and image-based cell counting of dissociated worms (*Baguñá and Romero, 1981*; *Thommen et al., 2019*). In this study, we sought to validate this quantification at the single-cell level in intact planarians. We calculated the neuron numbers and cell numbers in planarians with different sizes, including neurons specifically located in the brain (*Figure 2D–F*). We observed a proportional increase in the total count of neuron cells with the overall size of the body, comprising approximately 10% of the total body cells when the length is shorter than 7 mm (*Figure 2D*). Dividing the planarians into 2–6 mm and 7–9 mm groups, we observed that the neuron number to cell number ratio is significantly higher in the 2–6 mm planarian group (*Figure 2— figure supplement 1F*). However, it is important to mention that the number of neurons in the brain exhibits a linear increase with overall neuron count (*Figure 2F*). Beyond this threshold, the proportion of neurons in the brain relative to the total cell population decreases (*Figure 2D*). Referring to the images, the decreased ratio in large planarians may be caused by the reduced density of neurons in the brain (*Figure 2E*). These findings provide evidence to support the previous prediction and consistency between different planarian species (*Baguñá and Romero, 1981*; *Emili et al., 2023*). Because the tail is proportionately longer in large than in small planarians, the allometric growth of the planarians can be one possibility for this decrease along with the increase in animal size. The phenomenon may also suggest the existence of a threshold in the increase of planarian neuron numbers, which may ultimately contribute to some physiological changes, such as planarian fission.

We further analyzed different neuron types to examine their correlation with the increase in body size. Within the five types of neurons, we noticed that GABAergic, serotonergic, dopaminergic, and octopaminergic neurons increase in linear to the total cell number (*Figure 2—figure supplement 1G–J*). These results suggest that the above observation of the non-linear dynamics between neuron and total cell number is not likely from the octopaminergic, GABAergic, dopaminergic, and serotonergic neurons. Since our neuron pool may not include glutamatergic, glycinergic, and peptidergic neurons, the non-linear dynamics may be from cholinergic neurons or other neurons not included in our staining. We further analyzed the octopaminergic neurons in the brain and branch regions (*Figure 2G*) and the GABAergic neurons in the ventral medial (VM) and the dorsal lateral (DL) regions (*Figure 2J*; *Nishimura et al., 2008*; *Currie et al., 2016*). By quantifying these two groups, we found that the number of octopaminergic neurons in the brain increased concurrently with the overall increase of octopaminergic neurons; in contrast, the number of octopaminergic neurons in the branch region did not show a noticeable increase (*Figure 2H*). To examine if the morphology of the brain changes according to the growth of the body size, our measurement showed that the range of the angle of the brain lobe remains stable around 17.12°–20.88° (*Figure 2I*). Similarly, the proportion of dorsal lateral GABAergic neurons increased relative to the total number of GABAergic neurons; in contrast, the increase rate of the VM region neurons was much higher than the rate of the DL region (*Figure 2K*). These findings indicate that octopaminergic and GABAergic neurons in different locations may be controlled by distinct mechanisms for their growth.

## Differential increase trends by neuron types during planarian regeneration

To comprehensively observe the dynamic changes of the neuron population during regeneration in *S. med*, an experiment was conducted using the tail fragments of 5–6 mm-long planarians by cutting their posterior tails into 2 mm fragments. Over a period of 14 days, daily fixation of planarian fragments was carried out. Four planarian fragments were analyzed at each time point (*Figure 3A*, *Figure 3—figure supplement 1A*). Similar to homeostasis, a consistent surface area-to-volume ratio was maintained in the regenerative processes (*Figure 3B*, *Figure 3—figure supplement 1B*). We further segmented the brain during regeneration and found that the brain size increased during the 14-day regeneration period (*Figure 3—figure supplement 1C*). To subsequently analyze each neuron type, probes such as *chat*, *gad*, *tbh*, *tph*, *th*, and PI were used to stain the regenerating fragments (*Figure 3—figure supplement 1E*). In total, 251 samples were analyzed for 3D tissue reconstruction and cell segmentation.

Previous studies have shown that the balance of cell numbers in planarians is influenced by cell proliferation, differentiation, and cell death during regeneration (*Eisenhoffer et al., 2008*; *Takeda et al., 2009*; *Arnold et al., 2019*; *Oviedo et al., 2003*; *Hill and Petersen, 2015*). Cell numbers were counted from 0 to 14 days post-amputation (dpa). Cholinergic and serotonergic neurons were present not only in the brain but also distributed across the body's superficial layers (*Figure 3C*, *Figure 3—figure supplement 1D*). The count of cholinergic neurons initially started at ~7000 and continued to increase throughout the entire 14-day regeneration period (*Figure 3D*). For serotonergic neurons, they showed a similar pattern to cholinergic neurons (*Figure 3E*). GABAergic, octopaminergic, and dopaminergic neurons began to appear around days 3 and 4. Subsequently, the number of these neurons increased and reached a plateau after day 10 (*Figure 3F*, *Figure 3—figure supplement 1D*). It was reported that neurons exhibit an increase phase and plateau phase during planarian regeneration in *Dugesia japonica* (*Takeda et al., 2009*). Our results showed the similar pattern of neuron regrowth with two distinct phases, including an initial increasing phase (0–10 dpa) followed by a plateau phase (10–14 dpa).

Due to the linear and non-linear dynamics between neuron number and cell number in homeostatic growth, we further examined the dynamics of cell growth during regeneration. The GABAergic neurons in the VM and DL regions showed patterning on 4 dpa (*Figure 3G*). The growth of DL and VM GABAergic neurons occurs proportionally during regeneration, in which the DL GABAergic neurons increase faster than the VM GABAergic neurons (*Figure 3H*). In contrast, the octopaminergic neurons in the brain and branch regions began to appear on 3 dpa (*Figure 3I*). Similarly, the number of octopaminergic neurons in the brain region increases proportionally, while those in the branch region continue to increase until reaching a number approximately from 15 to 20 neurons at 13 or 14 dpa (*Figure 3J*). Moreover, the angle of octopaminergic neurons in the brain decreased during regeneration and stabilized at 20°, which remained consistent during homeostasis (*Figure 3—figure supplement 1E*, *Figure 2I*). These findings suggest that the reconstruction of DL, VM, branch, and main brain regions in planarians initiates concomitantly with the appearance of GABAergic and octopaminergic neurons. We further compared the ratio of different neuron types between planarians of the same body size at 14 dpa and in homeostasis. Our analysis revealed that the ratio of cholinergic and serotonergic neurons remained relatively constant in homeostasis. Conversely, the ratio of GABAergic, octopaminergic, and dopaminergic neurons is significantly lower in regeneration than in homeostatic planarians (*Figure 3K*). Different populations of neurons exhibit diverse growth patterns during the process of regeneration (*Takeda et al., 2009*). Our results provided additional evidence, obtained through comprehensive analyses of entire animals at the single-cellular level and in greater sample sizes, to support the model that proposes distinctive growth patterns for different populations of neurons during regeneration.

## Fine network of planarian musculature and distinct intersections at head-tail poles

Motivated by the crucial function of muscle in regeneration and the need to comprehend the control of movement by the neuromuscular system, we investigated the interaction between the neuronal and muscular systems. Initially, we examined the distribution of musculature in planarians. The 6G10 antibody was used to visualize the distribution of muscles throughout the planarian (*Ross et al., 2015*;

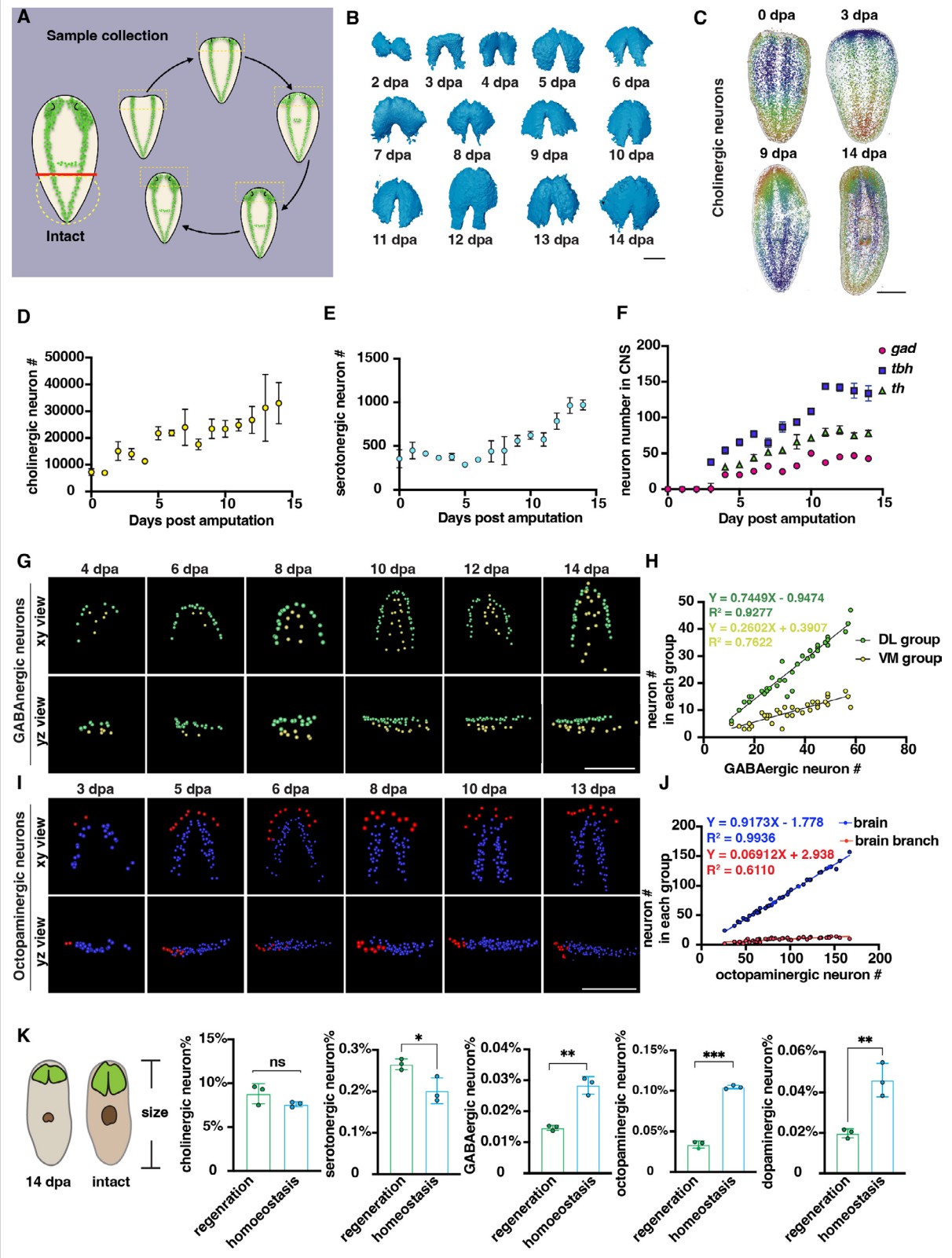

**Figure 3.** Changes in the number of planarian neurons during regeneration. (**A**) Workflow of sample collection of regenerative planarians. (**B**) Reconstructed 3D brains segmented from PI-labeled regenerative tail fragments at various time points post-amputation. Scale bar, 900 μm. (**C**) Representative xy views of segmented cholinergic neurons on 0-, 3-, 9-, and 14 days post-amputation. Scale bar, 1500 μm. (**D**) Dot plot shows the increase of cholinergic neurons (*chat*) at different time points after amputation. n≥3 in each condition. The data is shown as the mean ± SEM. (**E**) Dot

*Figure 3 continued on next page*

*Figure 3 continued*

plot shows the increase of serotonergic neurons (*tph*) at different time points after amputation. n≥3 in each condition. The data is shown as the mean ± SEM. (**F**) Dot plot shows the increase of GABAergic (*gad*), octopaminergic (*tbh*), and dopaminergic (*th*) neurons at different time points after amputation. n≥3 in each condition. The data is shown as the mean ± SEM. (**G**) Representative xy and yz views of segmented GABAergic neurons on 4, 6, 8, 10, 12, and 14 days post-amputation. Scale bar, 900 μm. (**H**) Dot plot shows the correlation of the total number of brain GABAergic neurons with the neuron number in the VM region (yellow dots) and DL region (green dots) in regenerative planarians. (**I**) Representative xy and yz views of segmented octopaminergic neurons on 3, 5, 6, 8, 10, 13 days post-amputation. Scale bar, 900 μm. (**J**) Correlation of the total number of the brain octopaminergic neurons with the octopaminergic neuron number in the brains (blue dots) and branches (red dots) in regenerative planarians. (**K**) Percentage of each type of neuron in the total cell number between 14 dpa planarians and the same size homeostatic planarians. n=3. Statistical significance was assessed by the two-tailed unpaired Student's t-test: **p<0.01, ***p<0.001; ns, not significant.

The online version of this article includes the following source data and figure supplement(s) for figure 3:

**Source data 1.** Source data for *Figure 3*.

**Figure supplement 1.** Changes in the number of planarian neurons during tail regeneration.

**Figure supplement 1—source data 1.** Source data for *Figure 3—figure supplement 1*.

*Cebrià, 2016*; *Cote et al., 2019*). Consequently, we validated that the body-wall musculature of adult planarians is composed of four layers of fibers, including circular, diagonal, longitudinal, dorsal-ventral (DV), and intestinal muscle fibers from the outmost to the innermost (*Figure 4A*, *Figure 4—figure supplement 1A-D and J*, *Figure 4—video 1*).

With our higher-resolution images, we conducted segmentation to gain a better understanding of the organization and orientation of the muscle fibers (*Figure 4B-H*, *Figure 4—figure supplement 1E-G, J, N*). A ground truth comparison of our automated muscle fiber segmentation with the original image was conducted to show the consistency (*Figure 4—figure supplement 2*). The planarian primarily relies on the movement of its cilia, which are mainly located on the ventral surface of the body (*Rink et al., 2009*). By closely examining the fiber structure, it becomes apparent that the circular muscle fibers dominate in all directions of the dorsal muscle wall (*Figure 4C*). However, the ventral body-wall muscles contain diagonal and longitudinal fibers at the tail region (*Figure 4C*). In contrast, the proportion of these fibers decreases in the dorsal muscle wall (*Figure 4C*). DV fibers are shorter compared with fibers in other orientations (*Figure 4F*, *Figure 4—figure supplement 1E-G*). We observe that these DV fibers have close contact with diagonal and longitudinal fibers (*Figure 4G, H*, *Figure 4—figure supplement 1G, M, N*).

Internal organs, such as the eyes and intestine, consist of a sophisticated distribution of muscle fibers (*Scimone et al., 2020*). We were able to visualize the intricate musculature in these organs with a resolution of 120 nm. The eyes of planarians contain short, sparsely distributed muscle fibers (*Figure 4—figure supplement 1H–J*, *Figure 4—video 2*). The pharynx, which serves as the feeding organ, is a muscular tube characterized by external and internal monostratified epithelia (*Figure 4—figure supplement 1K, L*). The intestine muscle fibers are located around the intestine, which are short and connected with small muscle fibers (*Figure 4—figure supplement 1H–J*). Furthermore, we investigated the connection between the dorsal epidermis and pharynx. We found that DV muscle fibers extend from the diagonal and circular layers of the dorsal body-wall muscles and connect with the longitudinal fibers of the pharynx (*Figure 4—figure supplement 1M, N*, *Figure 4—video 3*). These observations suggest that the planarian musculature is an interconnected unit, with the internal tissue muscles connected to the external body-wall muscles. It is noted that previous studies reported that 6G10 does not label all body wall muscles equivalently with the limitation of predominantly labeling circular and diagonal fibers (*Scimone et al., 2017*; *Ross et al., 2015*). Our observation may be limited by this preference.

Through the 3D tissue reconstruction method, we validated that the dorsal and ventral muscle fibers combine with circular muscle fibers, resembling a cobweb-like structure in the anterior pole (*Cebrià, 2016*; *Li et al., 2019*). Moreover, we observed that the integration of the ventral and dorsal body wall muscles differs in the anterior and posterior regions of the body. In the posterior region, the dorsal and ventral muscle walls integrate differently with longitudinal muscle fibers (*Figure 4I*, *Figure 4—videos 4 and 5*). It raised the possibility of whether the different morphologies are related to the A-P polarity. We thus examined the muscle structure at both the anterior and posterior heads of *β-catenin-1* RNAi planarians. Both the anterior and posterior muscle fibers of *β-catenin-1* RNAi planarians resemble the cobweb-like structure (*Figure 4J*). These detailed structures suggest a correlation

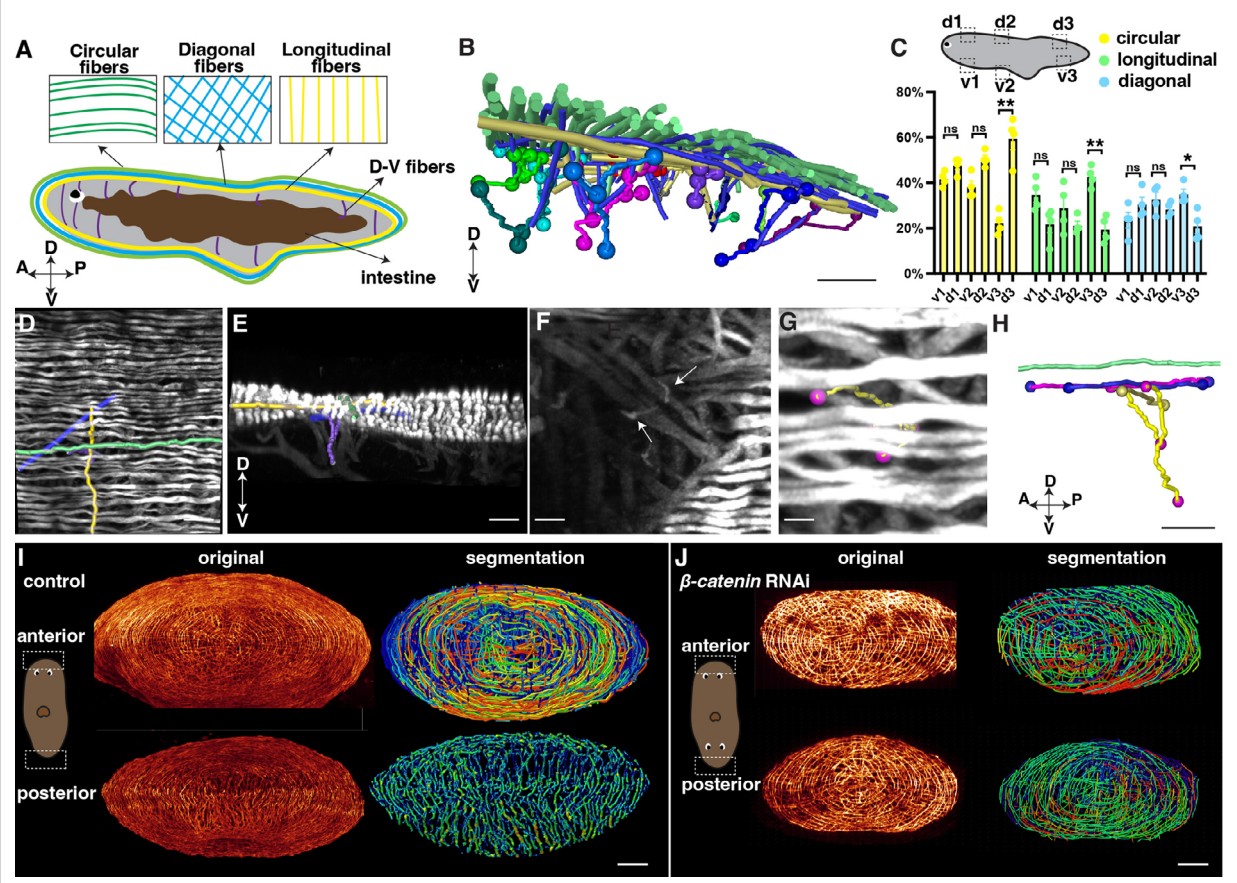

**Figure 4.** Distribution of planarian body-wall muscle fibers and their connections at the anterior and posterior poles. (**A**) Illustration of the major five muscle fibers in wild-type planarians according to their orientation and distribution. A: anterior; P: posterior; D: dorsal; V: ventral. (**B**) Segmented fibers of planarian body-wall muscle. Scale bar, 200 μm. (**C**) Schematic depicting selected segmented areas of planarian body-wall muscle, with a chart depicting the number of different orientational fibers in those regions. d: dorsal region; v: ventral region. The data is shown as the mean ± SEM. n≥3. Statistical significance was evaluated using the two-tailed unpaired Student's t-test, with *p<0.05, **p<0.01, ***p<0.001 indicating significance, while ns indicates lack of significance. (**D**) Planarian dorsal body-wall muscle fiber labeled with 6G10 antibody staining, with segmented circular (green), diagonal (blue), longitudinal (yellow), and D-V fibers (purple). Scale bar, 150 μm. (**E**) An xz view of the image in panel D. Scale bar, 150 μm. (**F**) Selected 100 μm depth region showing DV fiber (White arrows) located around diagonal fibers. Scale bar, 80 μm. (**G**) Body-wall muscle fiber of a region in the image of panel F. Segmented D-V fibers are shown in a tracked line. Scale bar, 50 μm. (**H**) An xz view of the segmented D-V fiber and its connecting fibers in panels F and G. Scale bar, 50 μm. (**I**) Xz projection of planarian anterior and posterior muscle fiber and their segmented muscle fibers in control planarian. Scale bar, 300 μm. (**J**) Xz projection of planarian anterior and posterior muscle fiber and their segmented muscle fibers in *β-catenin-1* RNAi planarian. Scale bar, 300 μm.

The online version of this article includes the following video, source data, and figure supplement(s) for figure 4:

**Source data 1.** Source data for *Figure 4*.

**Figure supplement 1.** Distribution of planarian muscle fibers.

**Figure supplement 2.** A comparison of automated muscle fiber segmentation with the original image.

**Figure 4—video 1.** 3D image of the body-wall muscle on the surface of a wild-type planarian.
https://elifesciences.org/articles/101103/figures#fig4video1

**Figure 4—video 2.** 3D view of muscle fibers surrounding the planarian eyes.
https://elifesciences.org/articles/101103/figures#fig4video2

**Figure 4—video 3.** 3D view of DV muscles connecting the body-wall muscles and the pharyngeal muscles.
https://elifesciences.org/articles/101103/figures#fig4video3

**Figure 4—video 4.** 3D view of muscle interconnection at the anterior pole of a wild-type planarian.
https://elifesciences.org/articles/101103/figures#fig4video4

**Figure 4—video 5.** 3D view of muscle interconnection at the posterior pole of a wild-type planarian.
https://elifesciences.org/articles/101103/figures#fig4video5

between muscle structure and the establishment of the anterior-posterior axis. The results highlight a noticeable contrast between the muscle fiber patterning of head and tail regions in terms of their responses to targets and adjustments in body posture. Unlike the tail, which doesn't need to react as actively, the head requires rapid reactions and precise posture changes. This is reflected in the more intricate muscle fiber arrangements observed in the head, suggesting a greater requirement for neural control.

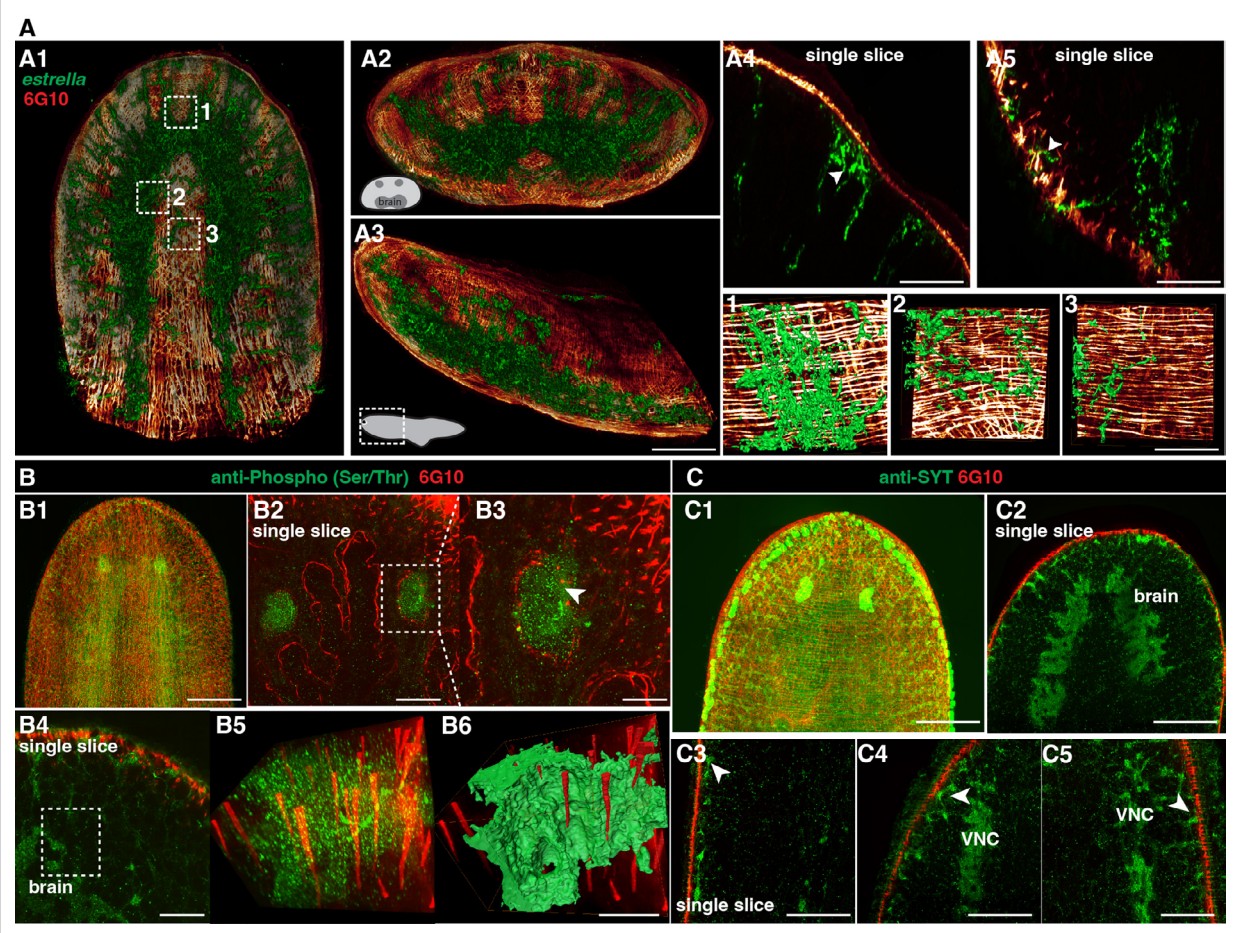

**Figure 5.** Spatial connection of planarian neurons and glial cells with muscle fibers. (**A1**) Dual staining for glial cells (*estrella*⁺) and muscles (6G10⁺) in a wild-type planarian. The representative brain region of xy is shown. (**A2**) Dual staining for glial cells (*estrella*⁺) and muscles (6G10⁺) in a wild-type planarian. The representative brain region of xz is shown. (**A3**) Dual staining for glial cells (*estrella*⁺) and muscles (6G10⁺) in a wild-type planarian. The representative brain region of yz is shown. Scale bar, 600 µm. (**A4**) A single slice of glial cells (*estrella*⁺) and muscles (6G10⁺) close to the epidermis of the anterior pole is shown. The arrowhead indicates the interaction of glial cells (*estrella*⁺) and muscles (6G10⁺). Scale bar, 250 µm. (**A5**) A single slice of glial cells (*estrella*⁺) and muscles (6G10⁺) close to the epidermis of the posterior pole is shown. The arrowhead indicates the interaction of glial cells (*estrella*⁺) and muscles (6G10⁺). Scale bar, 250 µm. (**1-3**) Zoomed in white dotted box region in A1 showing the *estrella*⁺ glial projection to the muscles. Scale bar, 150 µm. (**B1**) Dual staining for CNS (Anti-Phospho (Ser/Thr) staining) and muscles (6G10⁺) in a wild-type planarian. Scale bar, 600 µm. (**B2**) Single slice of anti-Phospho (Ser/Thr) and 6G10 expression around planarian's eye. Scale bar, 300 µm. (**B3**) Magnification of selected region in panel B2. Scale bar, 200 µm. (**B4**) Single slice of Anti-Phospho (Ser/Thr) and 6G10 staining in a brain region. Scale bar, 200 µm. (**B5**) Volume rendering of selected region in panel B4. (**B6**) Reconstruction of B5. Scale bar, 60 µm. (**C1**) Dual staining for CNS (anti-SYT staining) and muscles (6G10⁺) in a wild-type planarian. Scale bar, 600 µm. (**C2**) Single slice of Anti-SYT and 6G10 staining in brain region. Scale bar, 600 µm. (**C3**) Single slice of Anti-SYT and 6G10 staining close to the epidermis in the middle part of the body. Scale bar, 450 µm. (**C4**) Single slice of Anti-SYT and 6G10 staining close to the epidermis in the anterior part of the body. Scale bar, 450 µm. (**C5**) Single slice of Anti-SYT and 6G10 staining close to the epidermis in the anterior part of the body. Scale bar, 450 µm.

The online version of this article includes the following video for figure 5:

**Figure 5—video 1.** 3D view of glial cell projection onto the body-wall muscles.

https://elifesciences.org/articles/101103/figures#fig5video1

## Neural-muscular connection in planarian homeostasis and regeneration

We next observed the interaction between neuronal and muscular networks. The *estrella*[+] glial cells are widespread (*Figure 5A1–A3*), and the glial cells extend from the planarian CNS to the body-wall muscle fibers (*Figure 5A4 and A5*). On closer examination of the epidermis region, we observed a tight association between glial cells and muscle fibers (*Figure 5 1-3*, *Figure 5—video 1*). We further investigated the neuronal and muscular connection through dual staining of 6G10 antibody (muscles) and anti-Phospho (Ser/Thr; *Figure 5B1–B3*) or anti-SYT (neurons) (*Figure 5C1 and C2*). Both dual-labeling revealed that neural cells are closely associated with muscle fibers with their projections (*Figure 5B4–B6 and C3–C5*).

In the context of planarian regeneration, the expression of positional control genes (PCGs) by muscles is vital for orchestrating the complex process of tissue regrowth (*Scimone et al., 2017*). During the regeneration process, DV muscle fibers reconnect at the wound site, with longitudinal fibers and other muscle types gradually restoring the structure at the anterior tip and later integrating with circular and diagonal fibers through small DV fiber branches (*Figure 4—figure supplement 1O1-O3*). By visualizing the dual-staining of cholinergic neurons and muscle fibers, we can observe that cholinergic neurons are closely located to muscle fibers from day 0 (*Figure 4—figure supplement 1P1*). We found that the appearance of newly regenerated diagonal and circular muscle fibers is located closely with cholinergic neurons (*Figure 4—figure supplement 1P2, P3*). These results suggest that the newly formed muscle fibers organize and connect potentially with a strong correlation with CNS.

## Muscular infrastructure may support as a scaffold for the neuron projection

To further investigate the functional relationship between neuronal and muscular networks, we utilized previously reported gene RNAi strategy (*Roberts-Galbraith, 2022*) that are likely to impact the structures of muscle. Insulin may influence the proper signaling of skeletal muscles and neoblasts (*Miller and Newmark, 2012*; *Lei et al., 2016*; *Sylow et al., 2021*). *Inr-1* RNAi animals exhibited locomotion defects (*Lei et al., 2016*) and also displayed a higher length-to-width ratio compared to control animals (*Figure 6—figure supplement 1A*, *Figure 6—video 1*), suggesting possible neuromuscular system abnormalities. The body wall muscle fiber distribution in *inr-1* RNAi and *β-catenin-1* RNAi planarians differs from *egfp* RNAi planarians (*Figure 6A and B*). By calculating the concentration of different orientations of muscle fibers in *inr-1* RNAi and *β-catenin-1* RNAi planarians in a 500×250 × 300 µm$^3$ region (*Figure 6C*), we noticed that *inr-1* RNAi planarian has more circular fibers in both dorsal and ventral regions; *β-catenin-1* RNAi planarian has more longitudinal fibers in dorsal regions (*Figure 6C*).

Furthermore, an examination of sub-cellular neuronal expression is conducted using FISH labeling to identify cholinergic, dopaminergic, serotonergic, octopaminergic, GABAergic neurons, and glial cells (*Figure 6—figure supplement 1B, C*). *Inr-1* RNAi planarians manifested fewer cholinergic neurons and less glial branching in the brain region (*Figure 6—figure supplement 1B, C*). The distribution of GABAergic neurons was disordered in *inr-1* RNAi planarians (*Figure 6—figure supplement 1B*). Enhancement of muscle fibers revealed a substantial increase in the concentration of circular muscle fibers in both the ventral and dorsal regions (*Figure 6A-C*, *Figure 6—figure supplement 1D*). These results imply that the decreased presence of neurons and unusual arrangement of fibers with varying orientations within the body muscle wall may lead to locomotion impairments in *inr-1* RNAi planarians.

Considering the interaction between glial and muscle cells, the localization of *estrella*[+] glial and muscle fibers is further investigated. By dual-staining of anti-Phospho (Ser/Thr) and 6G10 in *inr-1* RNAi and *β-catenin-1* RNAi planarians, we found that the morphologies of neurons are normal, and they have close contact with muscle fibers (*Figure 6D and E*). However, by dual staining of *estrella* and 6G10, we found that the structure of glial cells is star-shaped in *egfp* RNAi planarian, however, glial cells in *inr-1* RNAi and *β-catenin-1* RNAi planarians have shorter cytoplasmic projections, and their sizes are smaller, lacking the major projection onto the muscles (*Figure 6D, E*, *Figure 6—figure supplement 1E-K*). Especially, in the posterior head of *β-catenin-1* RNAi planarians, the glial cell has few processes and can hardly connect with muscle fibers (*Figure 6E*). These results indicated that proper neuronal guidance and muscle fiber distribution could potentially contribute to facilitating accurate glial-to-muscle projections. Further investigation is required to distinguish the cell-autonomous and non-autonomous effects of *inr-1* RNAi and *β-catenin-1* RNAi on muscle and glial cells.

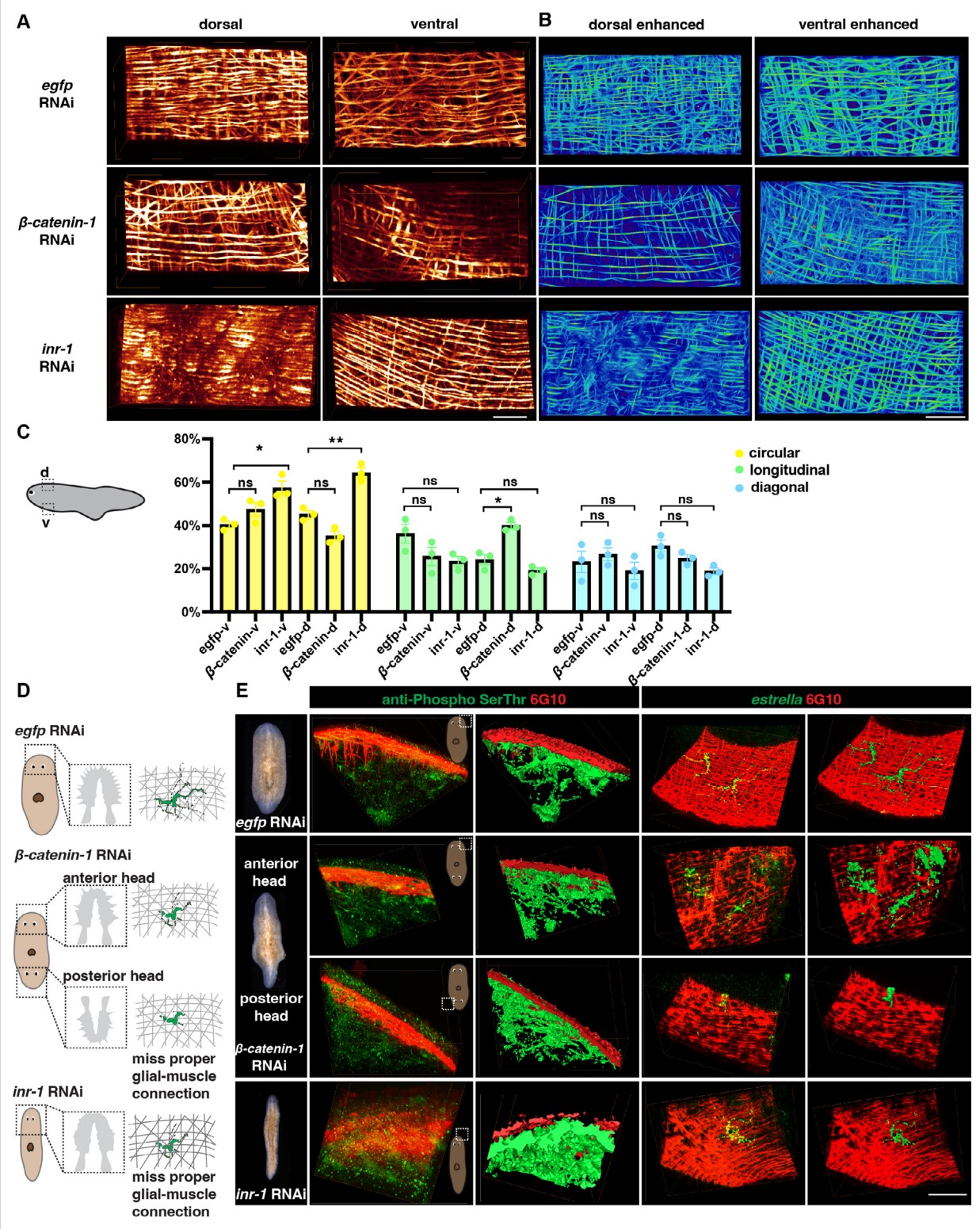

**Figure 6.** Neuron and muscle defects in *inr-1* RNAi and *β-catenin-1* RNAi planarians. (**A**) Representative images of dorsal and ventral muscle in *egfp* RNAi, *inr-1* RNAi, and *β-catenin-1* RNAi planarians. Scale bar, 120 μm. (**B**) Images depicting enhanced muscle fibers within dorsal and ventral regions of *egfp* RNAi, *inr-1* RNAi, and *β-catenin-1* RNAi planarian. Scale bar, 40 μm. (**C**) Schematic depicting selected segmented areas of planarian body-wall muscle in DV view. The plot shows the proportion of circular, longitudinal, and diagonal fibers in ventral and dorsal body muscle wall in *egfp* RNAi, *inr-1* RNAi, and *β-catenin-1* RNAi planarians. The data is shown as the mean ± SEM. n=3. Statistical significance was evaluated using the two-tailed unpaired Student's t-test, with *p<0.05, **p<0.01, ***p<0.001 indicating significance, while ns indicates lack of significance. (**D**) Illustration of *egfp* RNAi, *inr-1*

*Figure 6 continued on next page*

*Figure 6 continued*

RNAi, and *β-catenin-1* RNAi planarian glial cell structure and its connection with muscle fibers. (**E**) Anti-Phospho (Ser/Thr) and 6G10 staining and its reconstructed data in selected regions of *egfp* RNAi, *inr-1* RNAi, and *β-catenin-1* RNAi planarians (left). The *estrella*⁺ glia and 6G10⁺ muscles and their reconstructed images in selected regions of *egfp* RNAi, *inr-1* RNAi, and *β-catenin-1* RNAi planarians (right). Scale bar, 150 μm.

The online version of this article includes the following video, source data, and figure supplement(s) for figure 6:

**Source data 1.** Source data for *Figure 6*.

**Figure supplement 1.** Muscle fiber distribution and *estrella*⁺ glia localization in *inr-1* RNAi animals.

**Figure supplement 1—source data 1.** Source data for *Figure 6—figure supplement 1*.

**Figure 6—video 1.** Swimming behavior of *inr-1* RNAi and *egfp* RNAi planarians.

https://elifesciences.org/articles/101103/figures#fig6video1

## Discussion

In this study, we employed TLSM and C-MAP to investigate the spatial organization in planarians at the single-cell level. This combination offers several key advantages over standard techniques. For example, it enables high-throughput imaging across entire organisms with a level of detail and speed that is not easily achieved using confocal methods. This approach allows us to investigate the planarian nervous system at multiple developmental and regenerative stages in a more comprehensive manner, capturing large-scale structures while preserving fine cellular details. The ability to rapidly image whole planarians in 3D with this resolution provides a more efficient workflow for studying complex biological processes. Above all, our findings provide valuable insights into the cellular composition and neuronal diversity of planarians, shedding light on their regenerative capabilities and the interactions between muscle fibers and neurons during the regeneration process. One of the key observations in our study was the development of a robust segmentation method that allowed for the accurate identification and characterization of individual cells throughout the planarian body. This segmentation method, combined with the application of tissue expansion techniques, provided an accessible approach to obtaining high-resolution spatial information and enabled us to obtain a comprehensive view of the cellular landscape. Through the application of a 3D tissue reconstruction method, we investigated the development and diversity of various neuron types, including cholinergic, GABAergic, octopaminergic, dopaminergic, and serotonergic neurons, at the single-cell level. In addition, we pay attention to the neural networks of the planarian visual system, and we validated that there are contralateral axon projections onto the brain with single axon tracing results. It should be noted that the current resolution for our segmentation may be limited when resolving fibers within densely packed regions of the nerve tracts. Our analysis unveiled the intricate distribution of neurons throughout the planarian nervous system, encompassing regions such as the brain, ventral nerve cords, optic, and pharyngeal nerve complex.

Notably, as the planarian's body size increased, we observed that all neuron subtypes exhibited growth alongside the body cells, consistent with previous reports (*Takeda et al., 2009*; *Arnold et al., 2019*; *Oviedo et al., 2003*). Multiple approaches were used and validated to study the planarian cell changes (*Thommen et al., 2019*; *Oviedo et al., 2003*; *Hill and Petersen, 2015*). It is worth noticing that until reaching a threshold, beyond which the proportion of neurons decreases. This intriguing observation suggests a correlation with planarian fission, where the reduction in neuron proportion may be associated with the division of the planarian into two separate individuals. Our findings suggest that different neuron populations have coinciding regeneration speeds, and even the same neuron population may separate into different regeneration groups. Further investigations into the molecular and cellular mechanisms underlying this phenomenon would provide deeper insights into the factors governing planarian fission and the regenerative capacities of these organisms.

An important aspect of our study was the exploration of the interaction between muscle fibers and neurons during the regeneration process. By examining the structure, location, and regeneration of muscle fibers, as well as their connections to cholinergic neurons and glial cells labeled with *estrella*, we discovered a close correlation between ventral muscle fibers in the inner epidermal layer and cholinergic neurons and glial cytoplasmic projections. This finding suggests that muscle may play a crucial role in guiding the regeneration of the planarian nervous system, laying the foundation for future investigations into the neuron and muscle regeneration dynamics. Furthermore, 3D tissue imaging offers several advantages for clinical research and the medical industry by enhancing

diagnostic accuracy through improved spatial resolution. Notably, techniques such as light sheet microscopy and tissue clearing have shown their utility in visualizing human tissues, as well as mouse tissues and various other model animals (*Chung et al., 2013*; *Liebmann et al., 2016*; *Liu et al., 2016*). The integration of two modalities, TLSM and C-MAP, allows for effective 3D imaging with a resolution range of 120 nm to 500 nm. We envision opportunities to expand our efforts to include additional research organisms, such as axolotls, hydra, and frogs, thereby broadening the scope of our research.

In conclusion, our study utilizing TLSM and C-MAP expansion techniques provides a comprehensive understanding of planarian spatial organization and cellular dynamics at the single-cell level. The development of a robust segmentation method, combined with the analysis of various neuron types and their relationship with muscle fibers, highlights the intricate interactions between different cell populations during planarian regeneration. These findings significantly contribute to our knowledge of regenerative biology and provide a foundation for future studies to understand similar processes in other organisms. Further investigations into the functional significance of the observed cellular dynamics and interactions will undoubtedly advance our understanding of planarian biology and regenerative mechanisms.

# Materials and methods

**Key resources table**

| Reagent type (species) or resource | Designation | Source or reference | Identifiers | Additional information |
|---|---|---|---|---|
| Strain, strain background (*Schmidtea mediterranea*) | *Schmidtea mediterranea*, asexual | CIW4 | | |
| Strain, strain background (*Escherichia coli*) | DH5a | SangonBiotech | B528413 | |
| Strain, strain background (*Escherichia coli*) | HT115 | Sangon Biotech | A338983 | |
| Antibody | Anti-Digoxigenin (DIG)-POD, sheep polyclonal | Roche | 11207733910 | 1:1000 |
| Antibody | Anti-Fluorescein-POD, sheep polyclonal | Roche | 11426346910 | 1:1000 |
| Antibody | 6G10, mouse monoclonal | DSHB | 6G10-2C7 | IF(1:1000) |
| Antibody | Anti-Arrestin, rabbit polyclonal | Gift from Takeshi Inuoe | | IF(1:500), |
| Antibody | Anti-SYT, rabbit polyclonal | Gift from Takeshi Inuoe | | IF(1:100) |
| Antibody | anti-Phospho (Ser/Thr) Phe antibody,, rabbit polyclonal | CST | 9631 S | IF(1:1000) |
| Chemical compound | urea | SangonBiotech | A600148 | |
| Chemical compound | N-butyl diethanolamine | TCL chemicals | #B0725 | |
| Chemical compound | Triton X-100 | SIGMA | T8787-250ml | |
| Chemical compound | Tween20 | SIGMA | P9416-100ml | |
| Chemical compound | methanol | SCR | 80080418 | |
| Chemical compound | Acrylamide | Sangon Biotech | A100341 | |
| Chemical compound | N,N-Dimethylacrylamide | SigmaAldrich | M7279 | |
| Chemical compound | Sodium acrylate | Macklin | S833838 | |
| Chemical compound | 2,2'-Azobis[2-(2-imidazolin-2-yl) propane] dihydrochloride (VA-044) | Rhawn | R008695 | |
| Chemical compound | Formaldehyde | SIGMA | F8775 | |
| Chemical compound | Heparin | SIGMA | H3149 | |
| Chemical compound | Torula Yeast RNA | SIGMA | R6625 | |
| Chemical compound | Western Blocking Reagent | Roche | 11921681001 | |
| Chemical compound | Horse Serum | hyclone | N/A | |
| Chemical compound | Dextran Sulfate | Sangon Biotech | A600160 | |

*Continued on next page*

*Continued*

| Reagent type (species) or resource | Designation | Source or reference | Identifiers | Additional information |
|---|---|---|---|---|
| Chemical compound | Maleic acid | aladdin | M108866 | |
| Chemical compound | NAC | SIGMA | A7250 | |
| Chemical compound | DAPI | Thermo Fisher Scientific | D3306 | |
| Chemical compound | DIG RNA Labeling Mix | Roche | 11277073910 | |
| Chemical compound | Fluorescein RNA Labeling Mix | Roche | 11685619910 | |
| Chemical compound | DNase (RQ1 rnase free DNase) | Promaga | PAM 6101 | |
| Recombinant DNA reagent | Phusion High-Fidelity DNA Polymerase | NEB | M0530L | |
| Recombinant DNA reagent | T7 RNA Polymerase | Promega | P207E | |
| Commercial assay kit | MicroSpin G-50 Columns | Cytiva | 27533002 | |
| Commercial assay kit | StarPrep Gel Extraction Kit | GenStar | D205-04 | |
| Commercial assay kit | FsatPure Plasmid Mini Kit | Vazyme | DC201-01 | |
| Software, algorithm | Amira 3D | Thermo Fisher Scientific | | v2023 |
| Other | hybridization oven | xingfen | FYY-3 | equipment |
| Other | Thermocycler | Analytik Jena | Biometra TRIO 48 | equipment |
| Other | Microscope Cameras | Leica | DFC7000 T | equipment |

## Planarian culture and amputation

*Schmidtea mediterranea* clonal asexual strain CIW4 animals were maintained in 1×Montjuïc salts (1.6 mmol/L NaCl, 1.0 mmol/L CaCl$_2$, 1.0 mmol/L MgSO$_4$, 0.1 mmol/L MgCl$_2$, 0.1 mmol/L KCl and 1.2 mmol/L NaHCO$_3$ prepared in Milli-Q water) at 20 °C as previously described (*Cebrià and Newmark, 2005*), and were fed with liver paste every 3 days. Intact animals (1–14 mm in length) were starved for at least 7 days before each experiment. The worms (5–6 mm long) were amputated into two sections: anterior fragment (including pharynx) and tail.

## In situ hybridization and immunostaining

Fluorescence in situ hybridization was performed as previously described (*Pearson et al., 2009*; *King and Newmark, 2013*). Intact and regeneration samples were treated with reduction solution (1% v/v NP-40, 0.5% w/v SDS, and 50 mM DTT in 1×PBS) for 10 min at 37 °C, except for worms within 3 days post-amputation, and all samples were bleach with Ryan King's Bleach (5% Formamide, 1.2% H$_2$O$_2$ in 0.5×SSC) for 2 hr. Riboprobes were synthesized as previously described (*King and Newmark, 2013*). The primers are as follows: *Smed-chat* (SMED30031525) forward primer 5'-CTTTGGCACTTC CGATAAAC-3', reverse primer 5'-CCATTTCTGTTGTCGATTGG-3'; *Smed-gad* (SMED30001003) forward primer 5'-TATCAAAATAGGTCAGGGCC-3', reverse primer 5'-AAACGCCGCCATCTAATTTC -3'; *Smed-tbh* (SMED30017498) forward primer 5'-TTGGTCTGTTGAACCGAATC-3', reverse primer 5'-AATCTCCCTCAAAAGAGTCG-3'; *Smed-th* (SMED30012000) forward primer 5'-CACCAGTCAGAA TTTCATCG-3', reverse primer 5'-TATCATGAAAACCCGGATGG-3'; *Smed-tph* (SMED30012020) forward primer 5'-ACCAGACGAGGAAGATTTTC-3', reverse primer 5'-GCAAGACCAGCTAAAA AGTC-3'; *Smed-estrella* (KY024338.1) forward primer 5'-CAAATGCTGAGAATACTGGC-3', reverse primer 5'-TCGGAGTAAGCATCGTTTAG-3'. Animals were incubated with probes labeled with DIG (1:500) for more than 18 hr at 56 °C. Anti-DIG-POD 1:1000 (Roche) was used in MABT containing 5% Horse Serum and 0.5% Roche Western Blocking Reagent. The antibody 6G10 (1:1000, DSHB) was used in PBSTB (PBSTx 0.1%+ 1% Bovine Serum Albumin [Jackson Immuno Research Laboratories]) for FISH. For anti-DIG-POD labeling, samples were incubated overnight at 4 °C and then developed with FITC-conjugated tyramide (1:2000) in borate buffer containing 0.006% H$_2$O$_2$ for 1 hr at room temperature. For dual staining with antibodies, the worms were overnight incubated at 4 °C with 6G10 (1:1000), followed by incubation with the secondary antibody Goat Anti-mouse IgG H&L (HRP) pre-adsorbed (1:1000 in PBSTx0.3%, Abcam) on the following day. Subsequently, the worms were

incubated with rhodamine-tyramide (1:5000) in borate buffer containing 0.006% $H_2O_2$ for 1 hr on the third day. The same procedure was repeated for the additional antibody staining, including anti-Arrestin (rabbit, 1:500), anti-SYT (rabbit, 1:1100), and anti-Phospho (Ser/Thr) (rabbit, 1:1000).

### Tissue clearing for planarians

Tissue clearing was performed following the CUBIC protocol (*Matsumoto et al., 2019*), with specific optimizations for the planarian sample. The tissue-clearing solution consisted of 15% urea, 10% N-butyl diethanolamine, 10% Triton X-100, and 65% deionized water (ddH$_2$O). The specimens were immersed in this solution and gently shaken at either room temperature or 37 °C.

The duration of tissue clearing varied depending on the size and starvation state of the planarians, ranging from 30 min to overnight. It is worth noting that excessively long tissue clearing can compromise the integrity of planarian tissues. Starved planarians measuring 2–3 mm should skip the tissue-clearing step and proceed directly to the expansion procedure. Conversely, tail fragments that have been amputated from a 6 mm planarian require the tissue-clearing step.

For planarians of different sizes and developmental stages, the tissue clearing time should be adjusted based on their starvation status. An extended period of starvation allows for a reduction in tissue clearing time.

### C-MAP for planarians

The planarian specimen should be washed with 0.01 M PBS for 30 min at room temperature, with gentle shaking to ensure thorough clearance. To prepare the monomer solution, the final concentrations of the components should be as follows: 30% Acrylamide (AA), 0.075% N, N-Dimethylacrylamide (BA), 10% Sodium acrylate (SA), and 0.5% 2,2'-Azobis[2-(2-imidazolin-2-yl)propane] dihydrochloride (VA-044) in 0.01 M PBS. It is important to store the monomer solution at 4 °C and use it within 7 days.

Next, the planarian specimen should be immersed in the monomer solution for 30 min at 4 °C. The length of monomer incubation may vary depending on the size of the planarians, ranging from 30 min to overnight. For planarian of 2 mm length, the monomer incubation time is 30 min.

To perform the gelation step, it is necessary to work on ice. To create a double-layer gel that prevents direct contact between the specimen and the mold surface, start by adding 200 μL of the monomer solution onto the cap of a 1.5 mL Eppendorf (EP) tube on ice. Make sure that no sample is included in this first layer. The polymerization is initiated by exposing the gel to ultraviolet (UV) light for approximately 5 s, resulting in the formation of a coagulated gel with a tacky surface for support.

Afterward, carefully pipette the planarian specimen and 250 μL of monomer solution onto the first gel layer. This second layer should be solidified using UV light for 30 s, with the light source positioned 15 cm away from the sample. It is important not to use UV light to check the sample's position until it is properly placed in the mold. Once the specimen is embedded in the gel, separate the gel containing the specimen from the EP tube cap using tweezers and transfer it to ddH$_2$O in a 10 cm Corning cell culture plate. The specimen should be stored at room temperature for 2 days, with the ddH$_2$O changed after overnight incubation. Gentle shaking can be applied to expedite the expansion process.

To further increase the expansion ratio, the monomer solution should have the following final concentrations: 30% Acrylamide (AA), 0.05% N, N-Dimethylacrylamide (BA), 10% Sodium acrylate (SA), and 0.5% 2,2'-Azobis[2-(2-imidazolin-2-yl)propane] dihydrochloride (VA-044) in 0.01 M PBS.

### Labeling of planarian nuclei

After an overnight expansion, the gel underwent a twofold increase in size. To achieve accurate results, carefully use a blade accompanied by an illuminating light to precisely section the gel. These incisions should be in accordance with the contour of the planarian specimen, resulting in a cuboid shape. Subsequently, immerse the trimmed gel once again in fresh ddH$_2$O supplemented with 0.50 μg/mL of Propidium Iodide (PI) for nuclei staining at room temperature with gentle shaking. Stain the planarian overnight and wash the sample with ddH$_2$O for 10 min before imaging.

### Sample mounting

The planarian specimen, which had been embedded within the gel, was carefully trimmed with a blade to achieve a flat bottom surface. Following this, the gel was affixed onto a thin magnet using adhesive

glue. The magnet's dimensions were modifiable to align with the sample's proportions. Lastly, the gel-magnet assembly was secured onto a designated sample holder prepared for imaging.

## Imaging

The configuration and operational details of the microscope were described in earlier publications (*Chen et al., 2020*; *Feng et al., 2021*). Employing distinct arrangements of light sheet configuration and detective objectives, the expanded planarian specimen was subjected to imaging for specific experiment purposes. The planarians labeled with nuclei and neuron pool were imaged with OLYMPUS MV PLAPO 1×objective with micron-scale spatial resolution. The planarians labeled with 6G10 and *estrella* were imaged with OLYMPUS 10×0.6 SV MP to achieve sub-micron spatial resolution. The resolution can be up to ~70 × 70 × 210 nm$^3$ with this combination of objective and tilling light sheets. The image processing, registration, and merging procedure was described in detail in a previous publication (*Chen et al., 2020*).

## Resolution calculation

For cellular resolution imaging, we utilized a 1×air objective with a numerical aperture (NA) of 0.25 and a working distance of 60 mm (OLYMPUS MV PLAPO). The voxel size used was 0.8×0.8 × 2.5 µm$^3$. This configuration resulted in a resolution of 2×2 × 5 µm$^3$ and a spatial resolution of 0.5×0.5 × 1.25 µm$^3$ with 4×isotropic expansion. Alternatively, for sub-cellular imaging, we employed a 10×0.6 SV MP water immersion objective with 0.8 NA and a working distance of 8 mm (OLYMPUS). The voxel size used in this configuration was 0.26×0.26 × 0.8 µm$^3$. As a result of this configuration, we achieved a resolution of 0.5×0.5 × 1.6 µm$^3$ and a spatial resolution of 0.12×0.12 × 0.4 µm$^3$ with a 4.5×isotropic expansion.

## RNAi interference

*egfp*, without nucleotide sequence homology in planarians, was used as control RNAi. Animals were fed 1–6 times bacterially colored food (90% liver, 5.5% water containing 1×Montjuïc salts with 4.5% red food coloring)-expressed *egfp*, *smed-inr-1*, and *β-catenin-1* double-stranded RNA, once every 3 days. Animals were fixed 7 days after the last feeding.

## Nuclei quantification

The cellular quantification workflow was developed using the Amira 3D software environment (https://www.thermofisher.com/software-em-3d-vis/customerportal/download-center/amira-avizo-3d-installers/). A recipe and an example for nuclei counting can be assessed from https://zenodo.org/records/11724834. This recipe is designed for the segmentation of planarian nuclei-labeled images. The image analysis modules in the recipe provide flexibility for users to interactively check results at each step. Modules such as interactive thresholding and structure enhancement filters are designated as check breakpoints for parameter adjustment. The workflow consists of the following steps:

1. Use the volume edit module to eliminate the noise and irrelevant signals in the background. Select the noise or overexposed small objects using the lasso tool or handle box.
2. Apply either the Anisotropic diffusion or Gaussian filter to process the volumetric data. In the volume edit module, divide the planarian into two parts (the head and the remaining regions) using the lasso too. Use the segmentation editor to visualize the original image and the selection. Annotate regions for addition or deletion using the brush tool, and speed up the process using the interpolate function. Check the selected data using xy, yz, xz views, and 3D renderings
3. Smooth the data use either a Gaussian filter or anisotropic diffusion filter. In this workflow, we use an additional built-in Python script to enhance the nuclei edges and remove the noise.
4. To further enhance the visualization of cellular structures and nucleus boundaries, a 3D structure enhancement filter module was implemented. The 3D Hessian ball recognition port was selected to match the circular structure while removing others. This step is set as a breakpoint in the recipe and the standard deviation port can be adjusted to match the data structure. In the workflow for the head region, the standard deviation min/max pixel parameters were set as 1 and 6, respectively, with the standard deviation step of 1. In other body parts, the standard deviation min/max pixel parameters were set as 1 and 3, respectively, with a standard deviation step of 1. The ball recognition in the 3D structure enhancement module was used to detect cell edges. Alternatively, the 2D Hessian tensor was selected for cell edges or background boundary

detection. The standard deviation min/max pixels ports were both set as 1, and the standard deviation step was set as 1.

5. Transform the grayscale image into a binary image using an interactive thresholding module. Adjust the threshold of binary pixels using an ortho slice of the original image and the binary signals until they match. This step is identified as a breakpoint in the recipe, and the threshold value can be adjusted on a case-by-case basis.
6. Apply the 'remove small spots' module to eliminate smaller objects of noise or background.
7. Use either the watershed segmentation or marker-based watershed segmentation modules to label individual neurons and demarcate their separation. Attach the segmentation function to label the results of cells and background. Check the segmentation results in the segmentation editor.
8. Finally, manual examination was performed slice by slice in segmentation editor to correct mis-segmentation or over-segmentation.

## Neuron quantification

The recipe of neuron counting and an example can be freely accessed from https://zenodo.org/records/11724834. The workflow used for quantifying neurons closely resembled that of nuclei quantification, with two important considerations. First, due to the sparser distribution of neurons compared to somatic cells, the overall neuron data can be analyzed without the need to distinguish between the brain and other regions. Second, adjustments were made to the minimum and maximum parameters of the structure enhancement function standard deviation based on the staining size of the neuronal markers. The workflow can be broken down into the following steps.

1. Anisotropic diffusion or Gaussian filter was employed initially to reduce the noise and smooth the data.
2. A structure enhancement filter was then used to enhance the neuron signals. This step is set as a breakpoint in the recipe, and the standard deviation port can be adjusted to match the data structure. In our example, the standard deviation was set to a minimum of 2 pixels and a maximum of 5 pixels, respectively, with a step size of 1.
3. Apply an interactive thresholding module to transform the grayscale volumetric image into a binary image. The threshold of binary pixels was adjusted using an ortho slice of the original image and the binary signals until they aligned. The interactive thresholding function is marked as a breakpoint in the recipe, and the threshold value can be adjusted on a case-by-case basis. The remove small spots module was then employed to remove small objects below a certain pixel value.
4. Finally, the labeling module was used to assign labels to each neuron. The labeling function was linked to cell and cell boundary results. The recipe was designed for use with neuron pool labeling datasets, and careful parameter adjustments at breakpoints are recommended.

## **Measurement of planarian length and volume**

The recipe and an example of measuring volumetric parameters can be freely accessed from https://zenodo.org/records/11724834. The workflow can be broken down into the following steps.

1. Due to the large data size, it is necessary is to apply the resampling function to reduce the volumetric planarian nuclei data below 1 GB to make it suitable for GPU computation.
2. Anisotropic diffusion should be implemented to decrease the sharp edges in both the signal and background of the planarian specimen, while preserving the external contour. To achieve this, it is recommended to increase the threshold of anisotropic diffusion and the number of iterations to smooth the data.
3. An interactive thresholding module should be executed to convert the grayscale image into a binary image. Adjust the threshold of binary pixels can be done by utilizing an ortho slice of the original image and the binary signals until they match.
4. After completing the transformation, it is crucial to apply a fill small holes module to mitigate any voids present within the particles.
5. To resolve uneven surface contours, the compute ambient occlusion algorithm module was utilized to calculate an ambient occlusion scalar field for the dataset. It is advised to increase the maximum distance value and the number of rays to further smooth the data.
6. The segmented portion and the padded region were integrated using an arithmetic algorithm.

7. Subsequently, a label analysis module, along with 3D measurements, was applied to determine notable metrics such as the length and volume of an expanded planarian.

## Neuron tracing

Single anti-Arrestin labelled neuron tracing analysis was conducted with Amira 3D filament editor. Initially, 10–50 single-layer images were selected for maximum intensity projection display, based on their grayscale signals. Next, we identified and selected the axons to be tracked by determining their starting and ending points. Subsequently, layers were selected for display using the maximum intensity projection method. This approach facilitated the segmental tracking of the main axon, as well as the identification of branching points, thereby enabling the comprehensive tracking of neural fibers in 3D. Following the completion of the fiber tracking, we conducted a validation process to assess the accuracy of the traced fibers by comparing them with the original single-layer images. This validation was performed in multiple directions, with different branches of the fibers displayed in distinct colors.

To render the anti-SYT, anti-Phospho (Ser/Thr), *estrella*, and anti-Arrestin labeled data, we utilized Amira 3D segmentation editor. First, the datasets were transformed from grayscale to binary using interactive thresholding. Subsequently, the mis-segmented areas were evaluated and corrected using either the brush tool or the lasso tool. Lastly, the data was displayed using either volume rendering or the surface module generation method.

## Muscle fiber tracing

The muscle fiber tracing workflow was developed using Amira 3D. The data and project file related to muscle fiber tracing can be freely accessed from https://zenodo.org/records/11724834. The workflow can be broken down into the following steps.

1. The fiber data set was initially processed with the unsharp 3D masking module. This module was used to enhance the clarity of fiber edges without introducing additional noise.
2. The structure enhancement filter was implemented to extract the inherent fiber characteristics. The standard deviation min/max pixels were both set to 1, with a standard deviation step of 1 in the rod recognition structure type.
3. After enhancement of the structure, we utilized the cylinder correlation module to identify the fiber location and orientation by specifying parameters such as the fiber length, inner diameter, and outer diameter of the target cylinder. For anterior planarian muscle segmentation scenario, we set cylinder length to 18, the angular sampling to 5, the mask cylinder radius to 3, and the outer cylinder radius to 2.8. Note that these parameters may vary with other datasets.
4. The fluorescent signals conforming to the cylindrical shape were isolated and enhanced using the cylinder correlation module. The trace correlation line module was then utilized to extract and locate the target fibers. The output from this module was visualized with a spatial graph, providing key parameters and detailed information about traced fiber length and orientation.

## Statistical analyses

Microsoft Excel and Prism 9 were used for statistical analysis. The data in all graphs are shown as the mean ± SEM. An unpaired two-tailed Student's $t$-test was used to determine the significance of differences between the two conditions. Differences for which $p<0.05$ were considered statistically significant.

## Acknowledgements

We thank all laboratory members for their constructive comments. KL is supported by the National Natural Science Foundation of China (32122032, 31970750), the "Pioneer" and "Leading Goose" R&D Program of Zhejiang (2024SSYS0030), Zhejiang Provincial Key Laboratory Construction Project, and the Westlake Education Foundation. LG is supported by the Zhejiang Province Natural Science Foundation (LR20C070002) and the Westlake Education Foundation.

## Additional information

### Funding

| Funder | Grant reference number | Author |
|---|---|---|
| National Natural Science Foundation of China | 32122032 | Kai Lei |
| National Natural Science Foundation of China | 31970750 | Kai Lei |
| The "Pioneer" and "Leading Goose" R&D Program of Zhejiang | 2024SSYS0030 | Kai Lei |
| Natural Science Foundation Project of Zhejiang | LR20C070002 | Liang Gao |

The funders had no role in study design, data collection and interpretation, or the decision to submit the work for publication.

### Author contributions

Jing Lu, Hao Xu, Conceptualization, Data curation, Formal analysis, Validation, Investigation, Visualization, Methodology, Writing – original draft, Writing – review and editing; Dongyue Wang, Yanlu Chen, Software, Methodology; Takeshi Inoue, Resources, Methodology, Writing – review and editing; Liang Gao, Kai Lei, Conceptualization, Resources, Data curation, Software, Formal analysis, Supervision, Funding acquisition, Validation, Investigation, Visualization, Methodology, Writing – original draft, Project administration, Writing – review and editing

### Author ORCIDs

Takeshi Inoue ⓘ https://orcid.org/0000-0003-3289-4478
Liang Gao ⓘ https://orcid.org/0000-0001-9983-0626
Kai Lei ⓘ https://orcid.org/0000-0003-0601-7391

Reviewer #1 (Public review): https://doi.org/10.7554/eLife.101103.3.sa1
Reviewer #3 (Public review): https://doi.org/10.7554/eLife.101103.3.sa2
Author response https://doi.org/10.7554/eLife.101103.3.sa3

## Additional files

### Supplementary files

MDAR checklist

### Data availability

Planarian cell counting recipe, planarian cell counting project file, planarian neuron counting recipe, planarian neuron counting project file, planarian allometry measure recipe, planarian allometry measure project file, planarian anterior muscle segmentation project file, planarian posterior muscle segmentation project file, and planarian nuclei data are available at https://doi.org/10.5281/zenodo.11724834. Planarian musculature data (labeled with 6G10) is availablle at https://doi.org/10.5281/zenodo.12533272. The raw data for statistical analysis in each figure have been provided in this paper as source data, which include source data of Figure 2, Figure 3, Figure 4, Figure 6, and Figure 1—figure supplement 1, Figure 2—figure supplement 1, Figure 3—figure supplement 1, Figure 6—figure supplement 1. The details of the protocols are described in the Materials and methods. Further information about the methodologies and resources are available upon request to the corresponding authors, Kai Lei (leikai@westlake.edu.cn) and Liang Gao (gaoliang@westlake.edu.cn).

The following datasets were generated:

| Author(s) | Year | Dataset title | Dataset URL | Database and Identifier |
|---|---|---|---|---|
| Jing L | 2024 | 3D Reconstruction of Neuronal Allometry and Neuromuscular Projections in Asexual Planarians Using Expansion Tiling Light Sheet Microscopy dataset1 | https://doi.org/10.5281/zenodo.11724834 | Zenodo, 10.5281/zenodo.11724834 |
| Jing L | 2024 | 3D Reconstruction of Neuronal Allometry and Neuromuscular Projections in Asexual Planarians Using Expansion Tiling Light Sheet Microscopy dataset2 | https://doi.org/10.5281/zenodo.12533272 | Zenodo, 10.5281/zenodo.12533272 |

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
