## [Editor Report · eLife Assessment]

Lu and colleagues developed an **important** imaging protocol that combines expansion microscopy, light-sheet microscopy, and image segmentation for use with the planarian *Schmidtea mediterranea*, a powerful model system for regeneration. This represents a substantial improvement on current standards and enables more rapid data acquisition. The utility of this **solid** protocol is demonstrated by quantifying several aspects of this flatworm's neural anatomy and musculature during homeostasis and regeneration. This work will be of interest to researchers looking to implement more systematic approaches towards imaging and quantifying intact specimens.

---

## [Referee Report · Reviewer #1 (Public review)]

Summary:

The planarian flatworm *Schmidtea mediterranea* is widely used as a model system for regeneration because of its remarkable ability to regenerate its entire body plan from very small fragments of tissue, including the complete and rapid regeneration of the CNS. Prior to this study, analysis of CNS regeneration in planaria has mostly been performed on a gross anatomical level. Lu et al. describe a careful and detailed analysis of the planarian neuroanatomy and musculature in both the homeostatic and regenerating contexts. To improve the effective resolution of their imaging, the authors optimized a tissue expansion protocol for planaria. Imaging was performed by light sheet microscopy, and the resulting optical sections were tiled to reconstruct whole worms. Labelled tissues and cells were then segmented to allow quantification of neurons, muscle fibers, and all cells in individual worms.

Strengths:

The resulting workflow can produce highly detailed and quantifiable 3D reconstructions at a rate that is fast enough to allow the analysis of large numbers of whole animals.

Weaknesses:

While Lu et al. have shown how their methodology and workflow can be used to image and quantify features from whole animals, it is unclear how well their technique as described will perform at sub-cellular resolutions based upon the data that they show.

---

## [Referee Report · Reviewer #3 (Public review)]

Summary:

In this manuscript, the authors apply tissue expansion and tiling light sheet microscopy to study allometric growth and regeneration in planaria. They developed image analysis pipelines to help them quantify different neuronal subtypes and muscles in planaria of different sizes and during regeneration. Among the strengths of this work, the authors provide beautiful images that show the potential of the approaches they are taking and their ability to quantify specific cell types in relatively large numbers of whole animal samples. Many of their findings confirm previous results in the literature, which helps validate the techniques and pipelines they have applied here. Among their new observations, they find that the body wall muscles at the anterior and posterior poles of the worm are organized differently and show that the muscle pattern in the posterior head of beta-catenin RNAi worms resembles the anterior muscle pattern. They also show that glial cell processes appear to be altered in beta-catenin or insulin receptor-1 RNAi worms. Weaknesses include some over-interpretation of the data and lack of consideration or citation of relevant previous literature, as discussed below.

Strengths:

This method of tissue expansion will be useful for researchers interested in studying this experimental animal. The authors provide high-quality images that show the utility of this technique. Their analysis pipeline permits them to quantify cell types in relatively large numbers of whole animal samples.

The authors provide convincing data on changes in total neurons and neuronal sub-types in different-sized planaria. They report differences in body wall muscle pattern between the anterior and posterior poles of the planaria, and that these differences are lost when a posterior head forms in beta-catenin RNAi planaria. They also find that glial cell projections are reduced in insulin receptor-1 RNAi planaria.

Comments on revisions:

The authors have satisfactorily addressed the major concerns of the previous reviewers.

---

## [Author Response]

The following is the authors’ response to the original reviews.

**Reviewer #1 (Public review):**
comment 1: Lu et al. use their workflow to visualize RNA expression of five enzymes that are each involved in the biosynthetic pathway of different neurotransmitters/modulators, namely chat (cholinergeric), gad (GABAergic), tbh (octopaminergic), th (dopaminergic), and tph (serotonergic). In this way, they generate an anatomical atlas of neurons that produce these molecules. Collectively these markers are referred to as the "neuronpool." They overstate when they write, "The combination of these five types of neurons constitutes a neuron pool that enables the labeling of all neurons throughout the entire body." This statement does not accurately represent the state of our knowledge about the diversity of neurons in *S. mediterranea*. There are several lines of evidence that support the presence of glutamatergic and glycinergic neurons, including the following. The glutamate receptor agonists NMDA and AMPA both produce seizure-like behaviors in S. mediterranea that are blocked by the application of glutamate receptor antagonists MK-801 and DNQX (which antagonize NMDA and AMPA glutamate receptors, respectively; Rawls et al., 2009). scRNA-Seq data indicates that neurons in S. mediterranea express a vesicular glutamate transporter, a kainite-type glutamate receptor, a glycine receptor, and a glycine transporter (Brunet Avalos and Sprecher, 2021; Wyss et al., 2022). Two AMPA glutamate receptors, GluR1 and GluR2, are known to be expressed in the CNS of another planarian species, D. japonica (Cebria et al., 2002). Likewise, there is abundant evidence for the presence of peptidergic neurons in S. mediterranea (Collins et al., 2010; Fraguas et al., 2012; Ong et al., 2016; Wyss et al., 2022; among others) and in D. japonica (Shimoyama et al., 2016). For these reasons, the authors should not assume that all neurons can be assayed using the five markers that they selected. The situation is made more complex by the fact that many neurons in *S. mediterranea* appear to produce more than one neurotransmitter/modulator/peptide (Brunet Avalos and Sprecher, 2021; Wyss et al., 2022), which is common among animals (Vaaga et al., 2014; Brunet Avalos and Sprecher, 2021). However the published literature indicates that there are substantial populations of glutamatergic, glycinergic, and peptidergic neurons in *S. mediterranea* that do not produce other classes of neurotransmission molecule (Brunet Avalos and Sprecher, 2021; Wyss et al., 2022). Thus it seems likely that the neuronpool will miss many neurons that only produce glutamate, glycine or a neuropeptide.

In response to your comments, we agree that our initial statement regarding the "neuron pool" overstated the extent of neuronal coverage provided by the five selected markers. We have revised the sentence as “The combination of these five types of neurons constitutes a neuron pool that enables the labeling of most of the neurons throughout the entire body, including the eyes, brain, and pharynx”.

Furthermore, we chose the five neurotransmitter systems (cholinergic, GABAergic, octopaminergic, dopaminergic, and serotonergic) based on their well-characterized roles in planarian neurobiology and the availability of reliable markers. However, we acknowledge the limitations of this approach and recognize that it does not encompass all neuron types, particularly those involved in glutamatergic, glycinergic, and peptidergic signaling, which have been documented in *S. mediterranea*. We have also added the content about other neuron types in our revised results section “Additionally, the neuron system of *S. mediterranea* is complex which characterized by considerable diversity among glutamatergic, glycinergic, and peptidergic neurons in planarians and many neurons in *S. mediterranea* express more than one neurotransmitter or neuropeptide, which adds further complexity to the system. We used five markers for a proof of concept illustration. By employing Fluorescence in Situ Hybridization (FISH), we successfully visualized a variety of planarian neurons, including cholinergic (*chat+*), serotonergic (*tph+*), octopaminergic (*tbh+*), GABAergic (*gad+*), and dopaminergic (*th+*) neurons based on their well-characterized roles in planarian neurobiology and the availability of reliable markers. (Figure S2A, Supplemental video 2) (Currie et al., 2016). The combination of these five types of neurons constitutes a neuron pool that enables the labeling of most of the neurons throughout the entire body, including the eyes, brain, and pharynx (Figure 1B).”

comment 2: The authors use their technique to image the neural network of the CNS using antibodies raised vs. Arrestin, Synaptotagmin, and phospho-Ser/Thr. They document examples of both contralateral and ipsilateral projections from the eyes to the brain in the optic chiasma (Figure 1C-F). These data all seem to be drawn from a single animal in which there appears to be a greater than normal number of nerve fiber defasciculatations. It isn't clear how well their technique works for fibers that remain within a nerve tract or the brain. The markers used to image neural networks are broadly expressed, and it's possible that most nerve fibers are too densely packed (even after expansion) to allow for image segmentation. The authors also show a close association between estrella-positive glial cells and nerve fibers in the optic chiasma.

Thank you for your detailed feedback. While we did not perform segmentation of all neuron fibers, we were able to segment more isolated fibers that were not densely packed within the neural tracts. We use 120 nm resolution to segment neurons along the three axes. Our data show the presence of both contralateral and ipsilateral projections of visual neurons. Although Figure 1C-F shows data from one planarian, we imaged three independent specimens to confirm the consistency of these observations. In the revised manuscript, we have included a discussion on the limitations of TLSM in reconstructing neural networks. In the discussion part, we added “It should be noted that the current resolution for our segmentation may be limited when resolving fibers within densely packed regions of the nerve tracts”.

comment 3: The authors count all cell types, neuron pool neurons, and neurons of each class assayed. They find that the cell number to body volume ratio remains stable during homeostasis (Figure S3C), and that the brain volume steadily increases with increasing body volume (Figure S3E). They also observe that the proportion of neurons to total body cells is higher in worms 2-6 mm in length than in worms 7-9 mm in length (Figure 2D, S3F). They find that the rate at which four classes of neurons (GABAergic, octopaminergic, dopaminergic, serotonergic) increase relative to the total body cell number is constant (Figure S3G-J). They write: "Since the pattern of cholinergic neurons is the major cell population in the brain, these results suggest that the above observation of the non-linear dynamics between neurons and cell numbers is likely from the cholinergic neurons." This conclusion should not be reached without first directly counting the number of cholinergic neurons and total body cells. Given that glutamatergic, glycinergic, and peptidergic neurons were not counted, it also remains possible that the non-linear dynamics are due (in part or in whole) to one or more of these populations.

We have revised the statement into “These results suggest that the above observation of the non-linear dynamics between neuron and total cell number is not likely from the octopaminergic, GABAergic, dopaminergic, and serotonergic neurons. Since our neuron pool may not include glutamatergic, glycinergic, and peptidergic neurons, the non-linear dynamics may be from cholinergic neurons or other neurons not included in our staining.”

**Reviewer #2 (Public review):**
Weaknesses:(1) The proprietary nature of the microscope, protected by a patent, limits the technical details provided, making the method hard to reproduce in other labs.

Thank you for your comment. We understand the importance of reproducibility and transparency in scientific research. We would like to point out that the detailed design and technical specifications of the TLSM are publicly available in our published work: Chen et al., Cell Reports, 2020. Additionally, the protocol for C-MAP, including the specific experimental steps, is comprehensively described in the methods section of this paper. We believe that these resources should provide sufficient information for other labs to replicate the method.

(2) The resolution of the analyses is mostly limited to the cellular level, which does not fully leverage the advantages of expansion microscopy. Previous applications of expansion microscopy have revealed finer nanostructures in the planarian nervous system (see Fan et al. Methods in Cell Biology 2021; Wang et al. eLife 2021). It is unclear whether the current protocol can achieve a comparable resolution.

Thank you for raising this important point. The strength of our C-MAP protocol lies in its fluorescence-protective nature and user convenience. Notably, the sample can be expanded up to 4.5-fold linearly without the need for heating or proteinase digestion, which helps preserve fluorescence signals. In addition, the entire expansion process can be completed within 48 hours. While our current analysis focused on cellular-level structures, our method can achieve comparable or better resolution and we will add this information in the revised manuscript as “It is important to point out that the strength of our C-MAP protocol lies in its fluorescence-protective nature and user convenience. Notably, the sample can be expanded up to 4.5-fold linearly without the need for heating or proteinase digestion, which helps preserve fluorescence signals. In addition, the entire expansion process can be completed within 48 hours. Based on our research requirement, two spatial resolutions were adopted to image expanded planarians, 2×2×5 μm^3^ and 0.5×0.5×1.6 μm^3^. The resolution can be further improved to 500 nm and 120 nm, respectively.”

(3) The data largely corroborate past observations, while the novel claims are insufficiently substantiated.A few major issues with the claims:Line 303-304: While 6G10 is a widely used antibody to label muscle fibers in the planarian, it doesn't uniformly mark all muscle types (Scimone at al. Nature 2017). For a more complete view of muscle fibers, it is important to use a combination of antibodies targeting different fiber types or a generic marker such as phalloidin. This raises fundamental concerns about all the conclusions drawn from Figures 4 and 6 about differences between various muscle types. Additionally, the authors should cite the original paper that developed the 6G10 antibody (Ross et al. BMC Developmental Biology 2015).

We appreciate the reviewer’s insightful comments and acknowledge that 6G10 does not uniformly label all muscle fiber types. We agree that this limitation should be recognized in the interpretation of our results. We have revised the manuscript to explicitly state the limitations of using 6G10 alone for muscle fiber labeling and highlight the need for additional markers. We have included the following statement in the Results section: “It is noted that previous studies reported that 6G10 does not label all body wall muscles equivalently with the limitation of predominantly labeling circular and diagonal fibers (Scimone et al., 2017; Ross et al., 2015). Our observation may be limited by this preference”. We would also clarify that the primary objective of our study was to demonstrate the application of our 3D tissue reconstruction method in addressing traditional research questions. Nonetheless, we agree that expanding the labeling strategy in future studies would allow for a more thorough investigation of muscle fiber diversity. Relevant citations have been properly revised and updated.

(4) Lines 371-379: The claim that DV muscles regenerate into longitudinal fibers lacks evidence. Furthermore, previous studies have shown that TFs specifying different muscle types (DV, circular, longitudinal, and intestinal) both during regeneration and homeostasis are completely different (Scimone et al., Nature 2017 and Scimone et al., Current Biology 2018). Single-cell RNAseq data further establishes the existence of divergent muscle progenitors giving rise to different muscle fibers. These observations directly contradict the authors' claim, which is only based on images of fixed samples at a coarse time resolution.

Thank you for your valuable feedback. Our intent was not to suggest that DV muscles regenerate into longitudinal fibers. Our observations focused on the wound site, where DV muscle fibers appear to reconnect, and longitudinal fibers, along with other muscle types, gradually regenerate to restore the structure of the injured area. We have revised the our statement as:“During the regeneration process, DV muscle fibers reconnect at the wound site, with longitudinal fibers and other muscle types gradually restoring the structure at the anterior tip and later integrating with circular and diagonal fibers through small DV fiber branches (Figure S5O1-O3).”

(5) Line 423: The manuscript lacks evidence to claim glia guide muscle fiber branching.

We agree with your concerns that our statement may be overestimated. We have removed this statement from the revised version. Instead, we focused on describing our observations of the connections between glial cells and muscle fibers. We have revised the section as follows: “Considering the interaction between glial and muscle cells, the localization of *estrella+* glia and muscle fibers is further investigated. By dual-staining of anti-Phospho (Ser/Thr) and 6G10 in *inr-1* RNAi and *β-catenin-1* RNAi planarians, we found that the morphologies of neurons are normal, and they have close contact with muscle fibers (Figure 6D, E). However, by dual staining of *estrella* and 6G10, we found that the structure of glial cells is star-shaped in *egfp* RNAi planarian, however, glial cells in *inr-1* RNAi and *β-catenin-1* RNAi planarians have shorter cytoplasmic projections, and their sizes are smaller, lacking the major projection onto the muscles (Figure 6D, E, Figure S6E-K). Especially, in the posterior head of *β-catenin-1* RNAi planarians, the glial cell has few axons and can hardly connect with muscle fibers (Figure 6E). These results indicated that proper neuronal guidance and muscle fiber distribution could potentially contribute to facilitating accurate glial-to-muscle projections.

(6) Lines 432/478: The conclusion about neuronal and muscle guidance on glial projections is similarly speculative, lacking functional evidence. It is possible that the morphological defects of estrella+ cells after bcat1 RNAi are caused by Wnt signaling directly acting on estrella+ cells independent of muscles or neurons.

We understand that this approach is insufficient and we have revised the this section as follows: “Further investigation is required to distinguish the cell-autonomous and non-autonomous effects of *inr-1* RNAi and *β-catenin-1* RNAi on muscle and glial cells.”

(7) Finally, several technical issues make the results difficult to interpret. For example, in line 125, cell boundaries appear to be determined using nucleus images; in line 136, the current resolution seems insufficient to reliably trace neural connections, at least based on the images presented.

We use two setups for imaging cells and neuron projections. For cellular resolution imaging, we utilized a 1× air objective with a numerical aperture (NA) of 0.25 and a working distance of 60 mm (OLYMPUS MV PLAPO). The voxel size used was 0.8×0.8×2.5 μm^3^. This configuration resulted in a resolution of 2×2×5 μm^3^ and a spatial resolution of 0.5×0.5×1.25 μm^3^ with 4.5× isotropic expansion. Alternatively, for sub-cellular imaging, we employed a 10×0.6 SV MP water immersion objective with 0.8 NA and a working distance of 8 mm (OLYMPUS). The voxel size used in this configuration was 0.26×0.26×0.8 μm^3^. As a result of this configuration, we achieved a resolution of 0.5×0.5×1.6 μm^3^ and a spatial resolution of 0.12×0.12×0.4 μm^3^ with a 4.5× isotropic expansion. The higher resolution achieved with sub-cellular imaging allows us to observe finer structures and trace neural connections.

Regarding your question about cell boundaries, we have revised the manuscript to specify that the boundaries we identified are those of each nucleus.

**Reviewer #3 (Public review)**:Weaknesses:(1) The work would have been strengthened by a more careful consideration of previous literature. Many papers directly relevant to this work were not cited. Such omissions do the authors a disservice because in some cases, they fail to consider relevant information that impacts the choice of reagents they have used or the conclusions they are drawing.For example, when describing the antibody they use to label muscles (monoclonal 6G10), they do not cite the paper that generated this reagent (Ross et al PMCID: PMC4307677), and instead, one of the papers they do cite (Cebria 2016) that does not mention this antibody. Ross et al reported that 6G10 does not label all body wall muscles equivalently, but rather "predominantly labels circular and diagonal fibers" (which is apparent in Figure S5A-D of the manuscript being reviewed here). For this reason, the authors of the paper showing different body wall muscle populations play different roles in body patterning (Scimone et al 2017, PMCID: PMC6263039, also not cited in this paper) used this monoclonal in combination with a polyclonal antibody to label all body wall muscle types. Because their "pan-muscle" reagent does not label all muscle types equivalently, it calls into question their quantification of the different body wall muscle populations throughout the manuscript. It does not help matters that their initial description of the body wall muscle types fails to mention the layer of thin (inner) longitudinal muscles between the circular and diagonal muscles (Cebria 2016 and citations therein).Ipsilateral and contralateral projections of the visual axons were beautifully shown by dye-tracing experiments (Okamoto et al 2005, PMID: 15930826). This paper should be cited when the authors report that they are corroborating the existence of ipsilateral and contralateral projections.

Thank you for your feedback. We have incorporated these citations and clarifications into the revised manuscript. We acknowledge the limitations of this approach and have added a statement for this limitation in the revised manuscript “It is noted that previous studies reported that 6G10 does not label all body wall muscles equivalently with the limitation of predominantly labeling circular and diagonal fibers (Scimone et al., 2017; Ross et al., 2015). Our observation may be limited by this preference.”

(2) The proportional decrease of neurons with growth in *S. mediterranea* was shown by counting different cell types in macerated planarians (Baguna and Romero, 1981; https://link.springer.com/article/10.1007/BF00026179) and earlier histological observations cited there. These results have also been validated by single-cell sequencing (Emili et al, bioRxiv 2023, https://www.biorxiv.org/content/10.1101/2023.11.01.565140v). Allometric growth of the planaria tail (the tail is proportionately longer in large vs small planaria) can explain this decrease in animal size. The authors never really discuss allometric growth in a way that would help readers unfamiliar with the system understand this.

Thank you for your feedback. We have incorporated these citations and clarifications into the revised manuscript “These findings provide evidence to support the previous prediction and consistency between different planarian species (Baguñà et al., 1981; Emili et al.,2023). Because the tail is proportionately longer in large than in small planarians, the allometric growth of the planarians can be one possibility for this decrease along with the increase in animal size. The phenomenon may also suggest the existence of a threshold in the increase of planarian neuron numbers, which may ultimately contribute to some physiological changes, such as planarian fission.”

(3) In some cases, the authors draw stronger conclusions than their results warrant. The authors claim that they are showing glial-muscle interactions, however, they do not provide any images of triple-stained samples labeling muscle, neurons, and glia, so it is impossible for the reader to judge whether the glial cells are interacting directly with body wall muscles or instead with the well-described submuscular nerve plexus. Their conclusion that neurons are unaffected by beta-cat or inr-1 RNAi based on anti-phospho-Ser/Thr staining (Fig. 6E) is unconvincing. They claim that during regeneration "DV muscles initially regenerate into longitudinal fibers at the anterior tip" (line 373). They provide no evidence for such switching of muscle cell types, so it is unclear why they say this.

We acknowledge that some of our conclusions were overclaimed given the current data, and we appreciate the opportunity to clarify and refine these claims in the revised manuscript. Due the technique reason, we have not achieved the triple-staining to address this concern. We hope to make a progress in our future studies. Regarding the statement that "DV muscles initially regenerate into longitudinal fibers at the anterior tip" (line 373), as addressed in our previous response, this statement was unclear. Our intent was not to imply that DV muscles switch into longitudinal fibers. Instead, we observed that muscle fibers reconnect at the wound site, with longitudinal fibers and other muscle types gradually restoring the structure. We have revised this section: “During the regeneration process, DV muscle fibers reconnect at the wound site, with longitudinal fibers and other muscle types gradually restoring the structure at the anterior tip and later integrating with circular and diagonal fibers through small DV fiber branches (Figure S5O1-O3).”

(4) The authors show how their automated workflow compares to manual counts using PI-stained specimens (Figure S1T). I may have missed it, but I do not recall seeing a similar ground truth comparison for their muscle fiber counting workflow. I mention this because the segmented image of the posterior muscles in Figure 4I seems to be missing the vast majority of circular fibers visible to the naked eye in the original image.

Thank you for raising this important point. We have included a ground truth comparison of our automated muscle fiber segmentation with the original image in the revised Figure S6. The original Figure S6 has been changed as Figure S7. Regarding the observation of missing circular fibers in Figure 4I, we agree that the segmentation appears to have missed a significant number of circular fibers in this particular image. This may have been due to limitations in the current parameters of the segmentation algorithm, especially in distinguishing fibers in regions of varying intensity or overlap.

(5) It is unclear why the abstract says, "We found the rate of neuron cell proliferation tends to lag..." (line 25). The authors did not measure proliferation in this work and neurons do not proliferate in planaria.

Thank you for pointing out this mistake. What we intended to convey was the increase in neuron number during homeostasis. We have revised the abstract “We found that the increase in neuron cell number tends to lag behind the rapid expansion of somatic cells during the later phase of homeostasis.”

(6) It is unclear what readers are to make of the measurements of brain lobe angles. Why is this a useful measurement and what does it tell us?

The measurement of brain lobe angles is intended to provide a quantitative assessment of the growth and morphological changes of the planarian brain during regeneration. Additionally, the relevance of brain lobe angles has been explored in previous studies, such as Arnold et al., Nature, 2016, further supporting its use as a meaningful parameter.

(7) The authors repeatedly say that this work lets them investigate planarians at the single-cell level, but they don't really make the case that they are seeing things that haven't already been described at the single-cell level using standard confocal microscopy.

Thank you for your comment. We agree that single-cell level imaging has been previously achieved in planarians using conventional confocal microscopy. However, our goal was to extend the application of expansion microscopy by combining C-MAP with tiling light sheet microscopy (TLSM), which allows for faster and high-resolution 3D imaging of whole-mount planarians. We have added in the discussion section: “This combination offers several key advantages over standard techniques. For example, it enables high-throughput imaging across entire organisms with a level of detail and speed that is not easily achieved using confocal methods. This approach allows us to investigate the planarian nervous system at multiple developmental and regenerative stages in a more comprehensive manner, capturing large-scale structures while preserving fine cellular details. The ability to rapidly image whole planarians in 3D with this resolution provides a more efficient workflow for studying complex biological processes.”